# Bounded contribution of human early visual cortex to the topographic anisotropy in spatial extent perception

Juhyoung Ryu [1] & Sang-Hun Lee[1✉]

To interact successfully with objects, it is crucial to accurately perceive their spatial extent, an enclosed region they occupy in space. Although the topographic representation of space in the early visual cortex (EVC) has been favored as a neural correlate of spatial extent perception, its exact nature and contribution to perception remain unclear. Here, we inspect the topographic representations of human individuals' EVC and perception in terms of how much their anisotropy is influenced by the orientation (*co-axiality*) and radial position (*radiality*) of stimuli. We report that while the anisotropy is influenced by both factors, its direction is primarily determined by radiality in EVC but by co-axiality in perception. Despite this mismatch, the individual differences in both radial and co-axial anisotropy are substantially shared between EVC and perception. Our findings suggest that spatial extent perception builds on EVC's spatial representation but requires an additional mechanism to transform its topographic bias.

---

[1] Department of Brain and Cognitive Sciences, Seoul National University, Seoul 08826, Republic of Korea. ✉email: visionsl@snu.ac.kr

An object occupies an enclosed region in space, which defines its spatial extent. For primates who manipulate objects with their hands, accurately estimating objects' spatial extents is crucial for successful adaptation to their surroundings[1]. For instance, to pick up the eggs differing subtly in shape, dexterous finger manipulation needs to be supported by accurate spatial extent estimation[2]. Attesting to the importance of spatial extent estimation, humans can discriminate subtle differences between objects' aspect ratios with exquisite finesse[3,4].

Given the fine precision of estimation performance and the topographic nature of spatial extent, the high-resolution spatial representation in the early visual cortex (EVC, V1/2/3) has been favored as a neural correlate of spatial extent perception[5,6]. Previous human neuroimaging[7–13] and monkey single-cell[14] studies have supported the critical roles of EVC in spatial extent perception, showing that the spatial representations of EVC are contextually modulated in a way matching those of perception.

In view of this evidence, it is expected that human perception of spatial extent will reflect the topographic anisotropies known to characterize the spatial representations of EVC or those formed at its preceding neural stages. Two such anisotropies have been reported in the literature: radial and co-axial anisotropies. Here, "anisotropy" refers to any topographic properties of neurons' receptive fields (RFs) or their local connections that lead to anisotropic representations of spatial extents of visual stimuli: "radial" when elongated along the radial axis on which the RF is positioned in retinotopic space, and "co-axial" when elongated along the axis aligned to the stimulus orientation in visual space. These anisotropies have been reported at multiple levels, including the shape of individual or population neurons' receptive fields (pRFs) and the spread of horizontal connections between neurons. Radial anisotropy has been found in the RFs of retinal ganglion cells in both animals[15–19] and humans[20] and in the pRFs of human V1[21,22]. On the other hand, co-axial anisotropy has been found in EVC's RFs[23,24] and long-range horizontal connections[25–27] in animals.

Although topographic anisotropy is a fundamental property of the spatial extent representation, it remains mostly unexplored whether and how the topographic anisotropies in EVC relate to those in perceived spatial extent. In this study, we concurrently characterized human individuals' pRFs in EVC and their perception of spatial extents for both types of topographic anisotropy and then thoroughly examined their relationship in multiple aspects. In doing so, we aimed to clarify the nature of the topographic anisotropy of EVC's spatial representation and its impact on the perception of spatial extent by addressing two significant yet unresolved issues.

First, it remains unclear which type of topographic anisotropy more strongly governs the spatial representation of EVC. The two aforementioned lines of evidence suggest that both types of anisotropy are likely to be present in the spatial representation of EVC. However, the strong correlation between the preferred orientation and the retinotopic radiality in individual neurons[28–31] or cortical sites[32–36] complicates the determination of which of the two anisotropies, radiality or co-axiality, is the dominant factor determining the topographic bias of EVC. Probing only one type of anisotropy without probing the other type cannot provide conclusive evidence regarding the presence or degree of the probed type of anisotropy. This is because both types of anisotropy can arise from the correlation between preferred orientation and retinotopic radiality (Fig. 1a, b). To address this indeterminacy, we conducted a functional magnetic resonance imaging (fMRI) experiment to examine the anisotropy in pRF spatial extent while manipulating the orientation of pRF-mapping stimuli such that the radial and co-axial anisotropies have distinct predictions about the pRF anisotropy (Fig. 1c–e; see

the detailed rationale for this solution in the first subsection of the Results). Taking the same approach, we conducted a psychophysics experiment to inspect both types of anisotropy in spatial extent perception. Thus, we could determine how much radiality and co-axiality exert their respective influences on EVC's pRF and spatial extent perception. This allowed us to evaluate the correspondence between EVC and perception in the relative predominance of radial and co-axial anisotropies.

Second, the topographic anisotropy in EVC has rarely been directly linked to that in perception at the individual level. We acknowledge a previous study[5] that compared the topographic anisotropy in V1 response spread to that in spatial extent perception. However, there are two limitations to this comparison: (i) it only examined co-axial anisotropy and not radial anisotropy, and (ii) the neural and behavioral data were obtained from different species, with monkeys providing the neural data and humans providing the behavioral data. We expanded on this research by collecting both neural and behavioral data from many (n = 27) human individuals and evaluating the respective degrees of radial and co-axial anisotropies in both domains. This approach allowed us to leverage individual differences to evaluate whether and how much the variability in EVC's topographic anisotropy relates to that of spatial extent perception. This variability-based evaluation, when combined with the correspondence-based evaluation described earlier, would provide insight into the functional roles of EVC in spatial extent perception within the visual information processing hierarchy. For instance, satisfying both evaluations would suggest that the information processing for spatial extent perception is largely completed as early as in EVC, whereas satisfying only the variability-based evaluation would suggest that spatial extent perception requires further information processing beyond EVC.

The correspondence analysis showed that, although radiality and co-axiality exert their respective influences on both EVC's pRF anisotropy and perceptual anisotropy, there was a clear mismatch in relative predominance between EVC and perception. The pRF anisotropy showed a radial bias, with a weak modulation by co-axiality. On the other hand, the perceptual anisotropy exhibited a co-axial bias, with a moderate modulation by radiality. Despite this mismatch in bias, the variability analysis showed that the inter-individual variability in the pRF anisotropy was substantially correlated with that in the perceptual anisotropy, for both types of anisotropy. These findings suggest that while EVC plays a significant role, its role is limited by implicating an additional mechanism in the downstream cortical regions beyond EVC, which receives high-fidelity information that is radially biased and then transforms it to match the co-axial bias in perceived spatial extent.

## Results

**The rationale for pRF estimation using radially and tangentially oriented gratings.** To assess how the two anisotropy factors, namely radiality and co-axiality, affect the spatial representation of EVC, we estimated the pRFs of individual voxels in EVC using visual stimuli with oriented texture. We used two types of gratings with different orientations in the retinotopic polar space: one oriented radially, and the other oriented tangentially (Fig. 1c).

The rationale behind our experimental design is grounded in two pieces of knowledge. The first concerns the relationship between the functional unit of EVC and the sampling unit of fMRI. The functional unit of EVC that is relevant to the purpose of the present work is the orientation hypercolumns. Based on the anatomy of V1[37–39], the sampling unit of fMRI ($2 \times 2 \times 2$ mm$^3$ voxel in our case) is likely to contain approximately 10–30

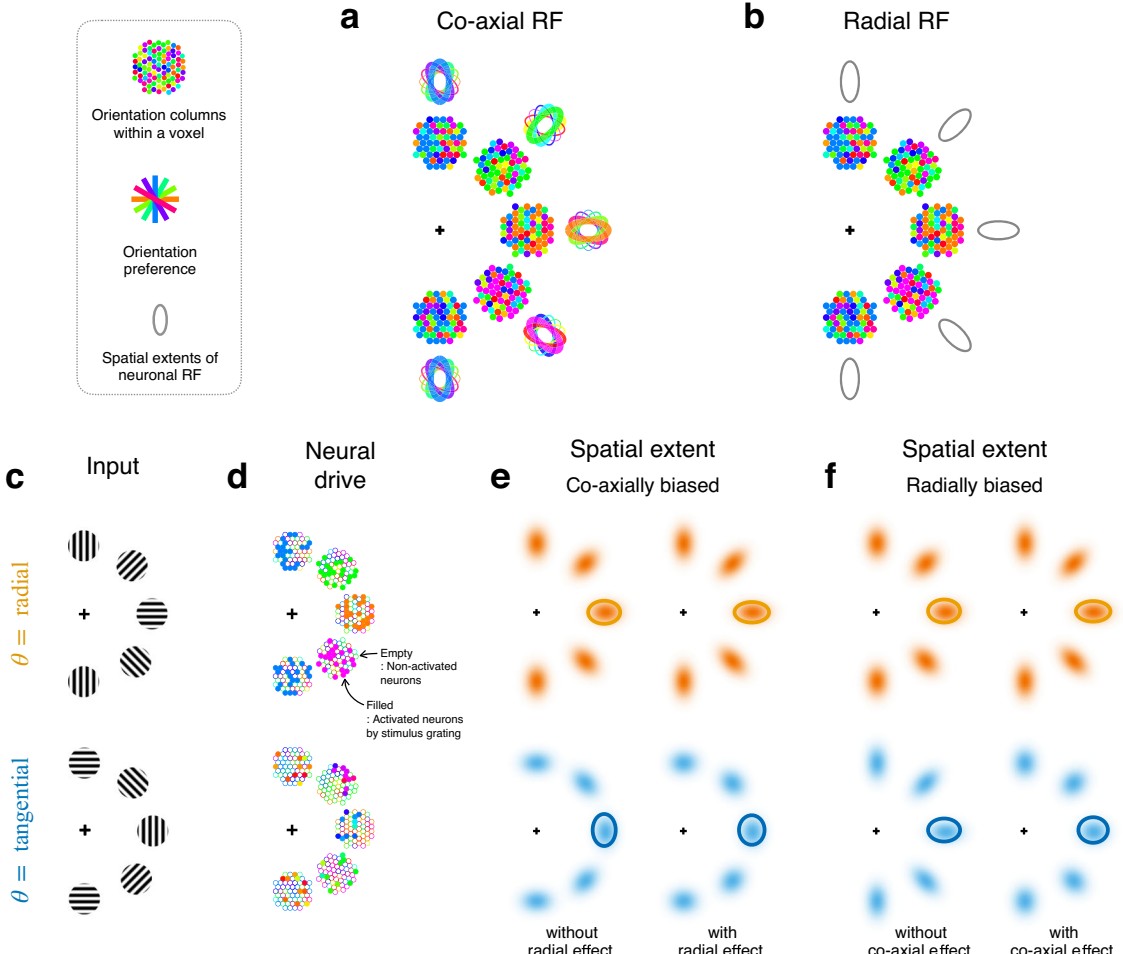

**Fig. 1 The rationale for pRF estimation using radially and tangentially oriented gratings to evaluate the respective influences of topographic factors on the spatial representation of EVC. a, b** Schematic illustrations of two hypothetical factors, co-axiality (**a**) and radiality (**b**), that govern the anisotropy of individual RFs of orientation-tuned neurons in the presence of a strong correlation between their orientation preference and polar-angle position. The fixation location is marked by crosshairs. Each cluster of colored dots represents orientation columns that are aggregated within a voxel representing a specific retinotopic position. The colors represent orientation preference, and their distributional biases reflect the strong correlation between columns' preferred orientation and polar-angle position. The ellipsoids surrounding the dot clusters represent the topographical anisotropy of neurons' RFs. If co-axiality governs, the RFs are elongated along their preferred orientation (**a**). If radiality governs, the RFs are elongated along the radial axis (**b**). **c** Visual inputs with radially (top) and tangentially (bottom) oriented gratings. **d** The neural drives by the gratings shown in **c**. **e, f** Illustrations of how the same neural drives shown in **d** induce different neural spatial extents depending on how strongly radiality and co-axiality govern the RF anisotropy. If only co-axiality influences the anisotropy, the neural spatial extent will be co-axially elongated, while the degree of co-axial elongation will not differ between the two orientation conditions (**e**, left). If co-axiality has a stronger influence on anisotropy than radiality, the neural spatial extent will be elongated in the co-axial direction, with a relatively high degree of elongation in the radial orientation condition (**e**, right). If only radiality influences the anisotropy, the neural spatial extent will be radially elongated, while the degree of radial elongation will not differ between the two orientation conditions (**f**, left). If radiality has a stronger influence on anisotropy than co-axiality, the neural spatial extent will be elongated in the radial direction, with a relatively high degree of elongation in the radial orientation condition (**f**, right).

hypercolumns. Therefore, the BOLD response of a single voxel to an arbitrary visual input can be understood as the response of a virtual hypercolumn to that input. The second concerns the correlation between the preferred orientations and polar-angle positions of single neurons[28–31] or local cortical sites[32–36]. From these two pieces of knowledge, we deduce that the relative proportions of orientation columns within the virtual hypercolumn vary systematically depending on the polar-angle position of a given voxel in retinotopic space. Specifically, the preferred orientation of the locally predominant column matches the polar-angle axis of the voxel (as depicted by the proportions of colored dots in Fig. 1a, b).

Critically, in the presence of this large-scale functional anisotropy in the orientation representation of EVC, the spatial extent of pRF is destined to be radially elongated, regardless of whether the dominant factor is radiality or co-axiality, as long as the pRF spatial extent is estimated using conventional, non-oriented stimuli (e.g., checkerboard pattern). When orientation-nonspecific stimuli are presented, all orientation columns are activated equally. However, the summed orientation preference of the neurons within a given voxel is determined by the preferred orientation of the orientation column aligned to the voxel's polar-angle position because that column, with radial orientation preference, is the dominant column within the voxel (as depicted by the proportions of color dots in Fig. 1a, b). Here, suppose the dominant factor is co-axiality, which means that all neurons' RFs are elongated along their preferred orientations (as depicted by the color-specific ellipsoids in Fig. 1a). Then, the aggregated

responses to orientation-nonspecific stimuli would result in radially elongated pRFs. Next, suppose the dominant factor is radiality. This means that all neurons' RFs are elongated along the radial axis, which is defined as the connecting line between the fovea and the position of the RF of a given voxel's position (as depicted by the gray ellipsoids in Fig. 1b). Then, the aggregated responses to orientation-nonspecific stimuli would result in radially elongated pRFs. Therefore, the estimated pRFs would display radial elongation regardless of whether the dominant factor in EVC's spatial representation is co-axiality or radiality as long as pRF mapping is done with orientation-nonspecific stimuli. This is a situation of degeneracy, wherein two causes result in the same consequence. As a result, we cannot draw any conclusion about which factor, radiality or co-axiality, dominantly affects the spatial representation of EVC from the spatial extent of pRFs estimated using the conventional, non-oriented stimuli.

To avoid this degeneracy, we estimated the pRFs by presenting gratings with radial (Fig. 1c, top) and tangential (Fig. 1c, bottom) orientations. This manipulation decorrelates the correlation between the preferred orientation and polar-angle position of visual neurons or their connections because the radial and tangential gratings would evoke the orientation-selective population within a voxel irrespectively of the preferred polar-angle position of the voxel (Fig. 1d). Importantly, this approach allows us to not only identify the governing factor in pRF spatial extent, which predominantly determines the biased axis of topographic anisotropy, but also assess whether and how much the remaining (non-governing) factor still exerts its modulatory influences on pRF spatial extent. Specifically, we will identify the dominant factor by inspecting the axis of elongation of pRF spatial extent in the tangential orientation condition: the elongation axis will be tangential if co-axiality governs the pRF anisotropy (Fig. 1e) and radial if radiality governs the pRF anisotropy (Fig. 1f). On the other hand, we can assess the absence or presence of the remaining factor's influence by comparing the degrees of elongation between the two stimulus orientation conditions: present if the degrees of elongation differ (left columns of Fig. 1e, f) and absent if the degrees of elongation do not differ (right columns of Fig. 1e, f). This is because the co-axiality and radiality factors are congruent in the radial orientation condition but incongruent in the tangential orientation condition.

**The pRF anisotropy of EVC is radially biased, with a weak modulation by co-axiality.** We acquired fMRI measurements from the EVCs of 29 human observers while they viewed a pack of Gabors that drifted slowly around a fixation spot. To investigate the anisotropy in pRF spatial extent along the radial versus tangential axes in the polar space, we presented spatially arranged Gabors in either a wedge or ring-shape, which traversed the visual field in the radial and tangential directions, respectively (Fig. 2a, b). By fitting a pRF model to the fMRI responses elicited by these stimuli at a single voxel, we estimated the voxel's pRF location (Fig. 2c) and its spatial extents along the radial and tangential axes (Fig. 2d).

With the rationale from the previous section (Fig. 1), we created two distinct viewing conditions. The Gabors were oriented radially in one condition and tangentially in the other condition (Fig. 2b). We then estimated pRFs separately for each of the two conditions. To effectively estimate the radial and tangential extents of pRFs, we modeled the pRF with the elliptical 2D Gaussian function with the axes fixed at the radial and tangential axes in the polar space. Then, for each voxel in each observer's EVC, we fitted this model to the voxel's fMRI time series in conjunction with the HIRF tailored to each individual's

EVC. Thus, we estimated two pRFs for each voxel, one defined with the radially oriented patterns and the other with the tangentially oriented patterns. For each of these two pRFs, we measured their anisotropy by comparing two spatial extents of the pRF model: one along the radial axis and the other along the tangential axis ($\sigma_{\text{radial−axis}}$ and $\sigma_{\text{tangential−axis}}$; the $\sigma$-defined contours of the pRF topography are shown for example voxel in Fig. 2d). We ensured the reliability of the pRF anisotropy estimate by checking the split-half reliability of the pRF parameter (Supplementary Fig. 1).

We first tested whether pRFs are anisotropic in the polar space by comparing the radial ($\sigma_{\text{radial−axis}}$) and tangential ($\sigma_{\text{tangential−axis}}$) spatial extents of the pRFs, separately for the two stimulus orientation conditions. In the radial orientation condition (Fig. 3a, left), the pRFs were elongated along the radial axis, significantly in all three areas of EVC (two-tailed paired $t$-tests; $t_{28} = 5.13$, $p = 1.9\text{e-}05$, 95% CI = [0.069°, 0.160°] for V1; $t_{28} = 2.09$, $p = 0.046$, 95% CI = [0.001°, 0.097°] for V2; $t_{28} = 8.24$, $p = 5.8\text{e-}09$, 95% CI = [0.193°, 0.321°] for V3). Likewise, in the tangential orientation condition (Fig. 3a, right), the pRFs were elongated along the radial axis, significantly in V1 and V3 ($t_{28} = 2.61$, $p = 0.014$, 95% CI = [0.012°, 0.101°] for V1; $t_{28} = 4.39$, $p = 1.5\text{e-}04$, 95% CI = [0.093°, 0.256°] for V3) and not significantly in V2 ($t_{28} = 1.48$, $p = 0.150$, 95% CI = [−0.011°, 0.067°]). These results indicate that radiality is the dominant factor governing the pRF anisotropy as the pRFs were observed to extend farther along the radial axis than the tangential axis, regardless of the stimulus orientation conditions.

Having identified radiality as the dominant factor, we then assessed the modulatory influence of co-axiality on the pRF anisotropy by comparing the degree of the radial elongation of pRF between the two orientation conditions. We quantified the degree of radial elongation with the radial bias index, $RI_{\text{pRF}}$, by taking the signed difference between the radial and tangential extents ($RI_{\text{pRF}} = \sigma_{\text{radial−axis}} - \sigma_{\text{tangential−axis}}$; see Materials and Methods subsection titled "Calculation of the pRF anisotropy based on subtractive contrast between the radial and tangential spatial extents of pRF" for the rationale behind our choice of subtractive contrast over divisive contrast). $RI_{\text{pRF}}$ tended to be smaller in the tangential orientation condition than in the radial orientation condition for all three areas of EVC (as indicated by the histograms in the upper-right corners of the panels in Fig. 3b and by the $RI_{\text{pRF}}$ values plotted against visual eccentricity in Supplementary Fig. 2) although these tendencies failed to reach the statistically significant level (two-tailed paired $t$-tests on $RI_{\text{pRF}}$s; $t_{28} = 1.64$, $p = 0.111$, 95% CI = [−0.014°, 0.130°] for V1; $t_{28} = 0.66$, $p = 0.517$, 95% CI = [−0.044°, 0.086°] for V2; $t_{28} = 1.49$, $p = 0.147$, 95% CI = [−0.031°, 0.197°] for V3). We obtained similar results using an alternative anisotropy measure based on divisive contrast ($t_{28} = 1.50$, $p = 0.146$, 95% CI = [−0.018°, 0.118°] for V1; $t_{28} = 0.37$, $p = 0.711$, 95% CI = [−0.051°, 0.074°] for V2; $t_{28} = 1.15$, $p = 0.261$, 95% CI = [−0.038°, 0.136°] for V3; Supplementary Fig. 3). These results indicate that EVC's pRF spatial extent is radially biased, with a weak modulation by co-axiality, albeit statistically insignificant.

**The anisotropy of perceived spatial extent is co-axially biased, with a moderate modulation by radiality.** To check the correspondence in topographic anisotropy between EVC and perception, we conducted a psychophysics experiment on the individuals who participated in the fMRI experiment. We measured the anisotropy in the perceived spatial extents of the gratings used in the fMRI experiment with a circularity-discrimination task[3,5]. On each trial, observers fixated on a

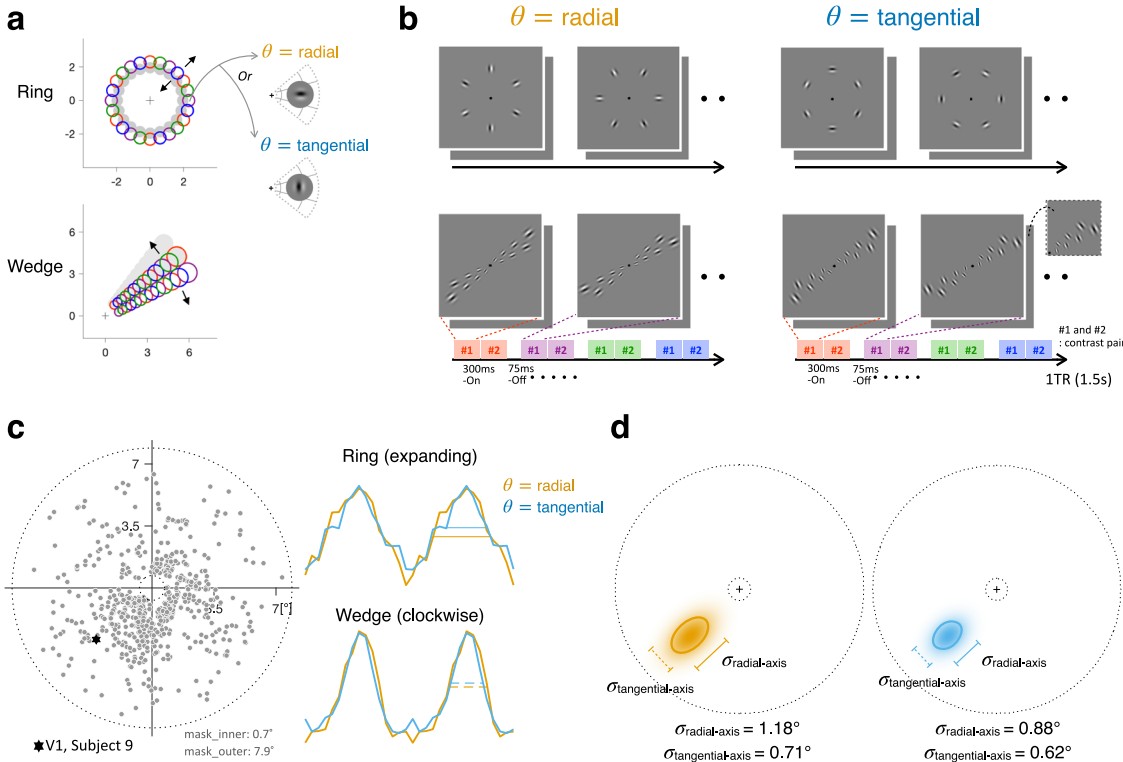

**Fig. 2 FMRI experimental design. a, b** Spatial arrangements (**a**) and temporal sequence (**b**) of pRF-mapping stimuli. At any given ring-shape or wedge-shape aperture, Gabor patches appeared within one of the four packs of ring-shaped or wedge-shaped apertures (circles with the same color, one color for one pack, shown in **a** in the fixed order (specified at the bottom of **b**). The ring-shape aperture traversed the visual field in the radial direction, either expanding or contracting (black arrows in **a**), whereas the wedge-shape aperture in the tangential direction, either clockwise or counter-clockwise). Each aperture remained at a fixed position for 1.5 s (1TR) and shifted by the size of the half-width of the aperture (gray shaded area in **a**) to make sure that the stimulus was turned on for 3 s at any given visual space. Within each pack of Gabor stimuli, we did not allow for spatial overlap between neighboring Gabors to minimize second-order interaction effects. **c, d** Estimation of pRF positions (**c**) and spatial extents (**d**). **c** Visual field coverage of V1, V2, and V3 voxels obtained from subject 9 and examples of 8-cycle-averaged fMRI time series from one single recording site (star mark) at V1 during the ring and wedge scans. Thin solid and dotted horizontal lines over the time series mark the full widths at half maximum (FWHMs) during the ring and wedge scans, respectively. The cycle-averaged fMRI time series was repeated twice for illustrative purposes. **d** Spatial extents of the best-fit pRFs of the example voxel is shown in **c**. Elliptical lines mark the pRF contour at half maximum; orange and blue colors indicate the radial and tangential orientation orientations, respectively.

fixation target and viewed a pair of Gabor stimuli presented at two mirrored positions on the radial axis from the fixation target. We asked them to choose the Gabor that looks closer to a perfect circle (Fig. 4a). The paired Gabors always had the same radial or tangential orientation but differed in their contrast envelope's physical (i.e., actual) spatial extent. The standard Gabor had an isotropic 2D Gaussian envelope with an aspect ratio of 1 ($AR = 1$), whereas the probe Gabor had an ellipsoidal envelope with a varying degree of aspect ratio ($AR = [0.5, 2]$). Here, the $AR$ is defined as $\sigma_{radial-axis}/\sigma_{tangential-axis}$ (Fig. 4b).

For each observer and each Gabor orientation condition, we derived a psychometric curve of the fraction of choosing the standard Gabor as a function of the $AR$ of the probe Gabor (Fig. 4c). We then estimated the $AR$ at which the probe Gabor appears as a perfect circle by identifying the minimum point on the curve (triangles in Fig. 4c). We will refer to this physical $AR$ defined at the minimum of the curve as probed $AR_{physical}$. By inverting probed $AR_{physical}$, we estimated the perceived $AR$ of the isotropic, standard Gabor. We will refer to this perceived $AR$ as $AR_{perc}$ and use its logarithmic value as the radial bias index of perceived spatial extent, $RI_{perc} = \log(AR_{perc}) = \log(\text{probed}\,AR_{physical}^{-1})$. Except for two observers, we were able to acquire psychophysical data and estimated $RI_{perc}$s from the observers whose pRF anisotropy was estimated.

As we had done for EVC's pRF, we assessed the influences of the two anisotropy factors on perceived spatial extent by comparing the mean values of $RI_{perc}$ between the two stimulus orientation conditions. We found that the mean $RI_{perc}$ was negative in the tangential orientation condition (two-tailed one-sample $t$-test; $t_{26} = -4.21$, $p = 2.7e-04$, 95% CI $= [-0.296, -0.102]$; histogram on the right of Fig. 5) and positive in the radial orientation condition ($t_{26} = 5.51$, $p = 8.8e-06$, 95% CI $= [0.164, 0.358]$; histogram at the bottom of Fig. 5). These results indicate that co-axiality predominantly governs the perceptual anisotropy by showing that the perceived spatial extent of the standard Gabor is extended farther along the tangential axis in the tangential orientation condition and along the radial axis in the radial orientation condition.

Having identified co-axiality as the dominant factor, we then assessed the modulatory influence of radiality on perceptual anisotropy by comparing the degree of the co-axial bias index ($CI_{perc}$) between the two orientation conditions. Here, $CI_{perc}$ measures the degree to which the perceived spatial extent is elongated along the axis of stimulus orientation. Thus, $CI_{perc}$ is identical to $RI_{perc}$ in the radial orientation condition ($CI_{perc;\theta=radial} = RI_{perc;\theta=radial}$) and can be obtained by $RI_{perc}$ with its sign inverted for the tangential condition ($CI_{perc;\theta=tangential} = -1 \times RI_{perc;\theta=tangential}$). $CI_{perc}$ tended to be

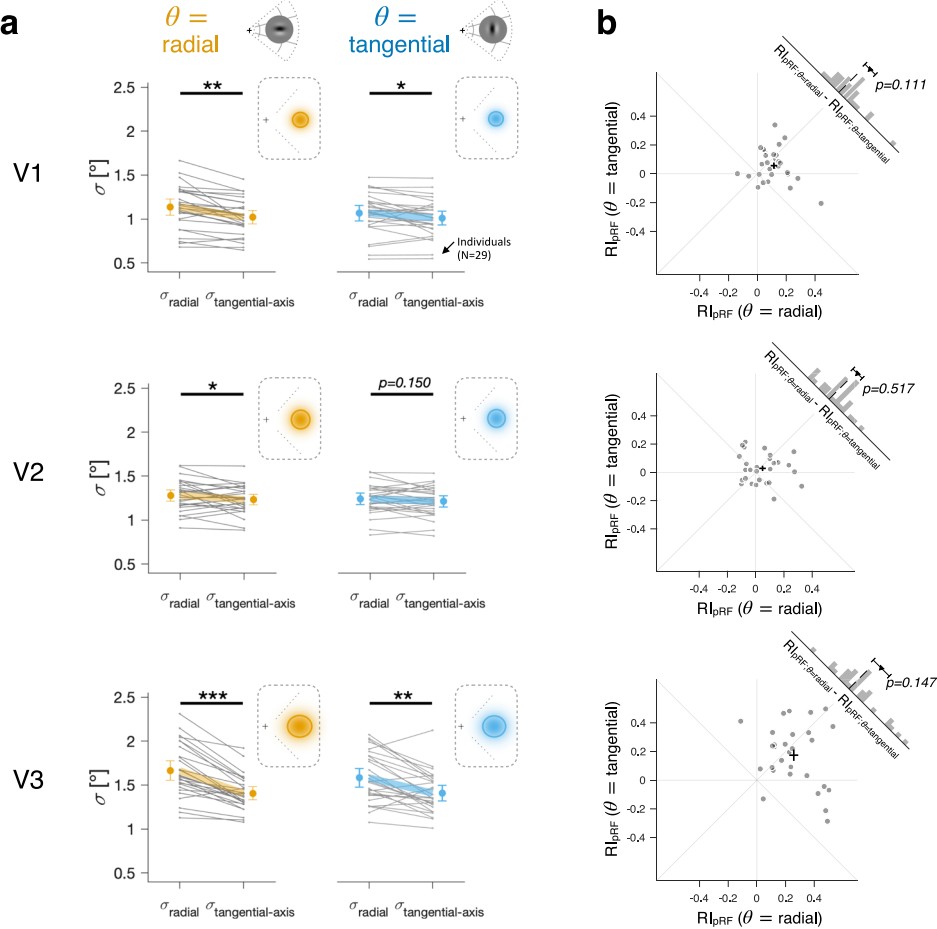

**Fig. 3 Topographic anisotropy in EVC's pRF. a** Comparison of the radial ($\sigma_{radial-axis}$) and tangential ($\sigma_{tangential-axis}$) extents of the pRFs defined by the radially oriented gratings (left) and by the tangentially oriented gratings (right). Insets show the across-individual averages of the pRF's spatial extents for the two stimulus orientation conditions. For illustrative purposes, the pRF topography is centered at the eccentricity of 3° on the horizontal meridian in the right visual field, where the horizontal and vertical widths of the ellipsoids mark the average widths of the pRFs at half maximum in the radial and tangential axes, respectively. Thin gray lines, 29 individuals; Thick colored lines and error bars, mean and 95% CI; Asterisks, statistical significance of two-tailed paired *t*-test; *$p < 0.05$, **$p < $ e-03, ***$p < $ e-08; Orange and blue colors, the radial and tangential orientation conditions. **b** Comparison of the radial bias indices of the pRF anisotropy between the radial, $RI_{pRF}(\theta = \text{radial})$, and tangential, $RI_{pRF}(\theta = \text{tangential})$, orientation conditions. The histogram in the upper-right corner of each panel shows the across-individual distribution of differences in $RI_{pRF}$s between the two orientation conditions. Dots, 29 individuals; Black crosshairs, across-individual means and their standard errors.

smaller in the tangential orientation condition than in the radial orientation condition (as indicated by the histograms in the upper-left corner of Fig. 5), which was statistically significant (two-tailed paired t-tests on $CI_{perc}$s; $t_{25} = 2.26$, $p = 0.033$, 95% CI = [0.006, 0.118]). These results indicate that the perceptual anisotropy is modulated by radiality while being predominantly governed by co-axiality.

**The pRF anisotropy and the perceptual anisotropy differ in bias but share the inter-individual variability.** The analyses conducted so far indicate that the topographic anisotropies of EVC and perception are opposite regarding which one of radiality and co-axiality is the dominant factor and which one is the modulatory one. As for the pRF anisotropy, its direction was biased by radiality, and its magnitude was modulated by co-axiality (as summarized by the solid horizontal bars in Fig. 6a). As for the perceptual anisotropy, its direction was biased by co-axiality, and its magnitude was modulated by radiality (as summarized by the empty vertical bars in Fig. 6a).

The substantial mismatch in anisotropy between EVC and perception appears at odds with the view that the neural

mechanism of spatial extent perception is largely completed as early as in EVC. This mismatch, when EVC's position along the hierarchical streams of information processing in the primate visual system was considered[40], suggests that there is a mechanism—presumably in the downstream cortical regions beyond EVC—that receives the high-fidelity yet radially biased input from EVC and topographically transforms it to display the co-axially biased anisotropy as observed in perception. This scenario of "hierarchical transformation" can explain why human spatial extent perception exhibits high-fidelity performance in aspect ratio discrimination[3,4] yet mismatches in anisotropy bias with EVC.

Under this scenario, any substantive variations in the topographic properties of EVC are likely to propagate to the downstream cortical regions, where spatial representations closely match those of perceived spatial extent. To check this possibility, we leveraged the substantial inter-individual differences in the impact of radiality and co-axiality on both the pRF (Fig. 3b) and perceptual (Fig. 5) anisotropies. Considering that the influences of radiality and co-axiality on anisotropy are not mutually exclusive but can coexist, as observed in the present work, we examined the

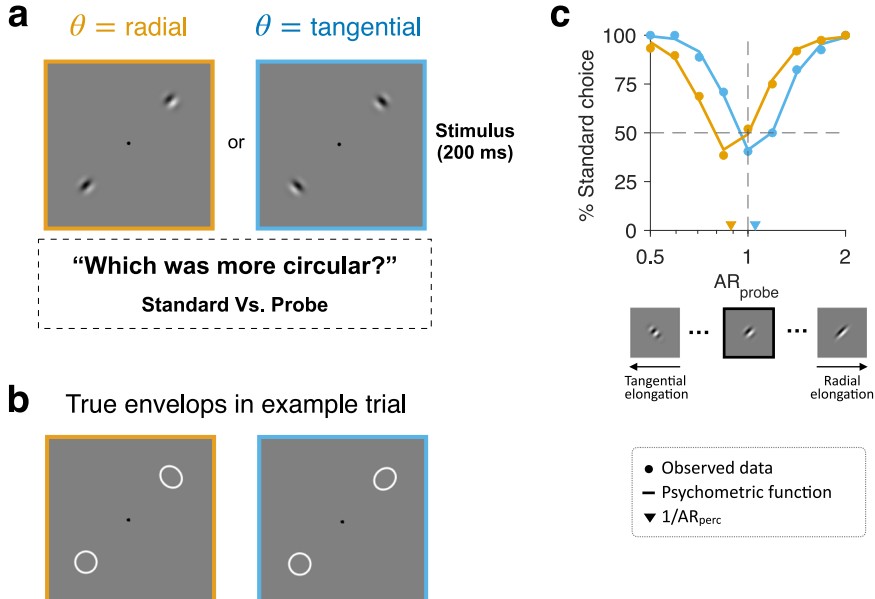

**Fig. 4 Circularity judgment task and example psychometric curves. a, b** Example pairs of Gabor stimuli (**a**) and their true spatial extents (**b**) under the radial (orange box) and tangential (blue box) orientation conditions. Each pair consisted of standard and probe Gabors, which always had the same radial or tangential orientation but differed in aspect ratio ($AR$, $\sigma_{radial-axis}/\sigma_{tangential-axis}$). The aspect ratio was fixed at 1 for the standard (lower left corners) and varied for the probe (upper-right corners). In each trial, observers briefly viewed the stimuli and selected the one appearing close to a perfect circle. **c** Example psychometric curves from a representative observer under the radial (orange) and tangential (blue) orientation conditions. The fraction of trials in which the standard stimulus was chosen is plotted as a function of the $AR$ of the probe ($AR_{probe}$) in a given trial. In the radial orientation condition, the observer's preference for the standard was lowest when the value of $AR_{probe}$ fell around a certain value smaller than 1 (as indicated by the orange downward triangle, which marks an estimate of the minimum point of the psychometric curve). In the tangential orientation condition, such a minimum point was found at the value of $AR_{probe}$ falling around a certain value slightly greater than 1 (as indicated by the blue downward triangle). By inverting these values of $AR_{probe}$ at the minima of the psychometric curves, the perceived aspect ratios ($AR_{perc}$s) were estimated. Dots, observed data; Solid lines, fitted psychometric functions; downward triangle, $1/AR_{perc}$; Inset images on the x-axis, the probe Gabors at three $AR$ values (0.5, 1, 2). Orange and blue colors, the radial and tangential orientation conditions.

inter-individual correlation between EVC and perception separately for their respective influences.

As for the radiality factor, we regressed the individuals' $RI_{perc}$ onto their across-area averages of $RI_{pRF}$ while holding the influence of the stimulus orientation. The regression was significant in both stimulus orientation conditions: across individuals, $RI_{perc}$ became increasingly positive as $RI_{pRF}$ became more positive, both in the radial (the orange dots in Fig. 6b; Pearson correlation $r$ and one-tailed $p$-value, $r_{(25)} = 0.52$, $p = 0.003$) and tangential (the blue dots in Fig. 6b; $r_{(25)} = 0.35$, $p = 0.039$) orientation conditions. These results indicate that the inter-individual variability in the radial bias in EVC's pRF anisotropy successfully relates to that in the corresponding radiality effect on the perceptual anisotropy.

As for the co-axiality factor, we need to regress the individual differences in the degree of co-axial bias in perceptual anisotropy onto that in the modulatory influence of co-axiality on the pRF anisotropy. Here, the regressor (i.e., the influence of co-axiality on the pRF anisotropy) was quantified by the co-axial modulation index ($CMI_{pRF}$), which is the subtraction of the $RI_{pRF}$ in the tangential orientation condition from the $RI_{pRF}$ in the radial orientation condition (as defined in the top panel of Fig. 6c, which corresponds to the differences in the $x$-axis values between the orange and blue dots paired across individuals in Fig. 6b). On the other hand, the regressand (i.e., the degree of co-axial bias in the perceptual anisotropy) was estimated by the co-axial bias index ($\overline{CI}_{perc}$), which is the average of the stimulus orientation-dependent co-axial bias indices, $CI_{perc;\theta=radial}$ and $CI_{perc;\theta=tangential}$, which was defined in the previous section (as defined in the

bottom panel of Fig. 6c). We found that the individuals' $\overline{CI}_{perc}$ values were significantly regressed onto their corresponding values of $CMI_{pRF}$ (Fig. 6d; Pearson correlation $r$ and one-tailed $p$-value, $r_{(25)} = 0.57$, $p = 0.001$). We observed similar results for the inter-individual variabilities of $CMI_{pRF}$ values in every area of EVC ($r_{(25)} = 0.40$, $p = 0.020$ for V1; $r_{(25)} = 0.42$, $p = 0.015$ for V2; $r_{(25)} = 0.57$, $p = 0.001$ for V3; Supplementary Fig. 4). These results indicate that the inter-individual variability in the modulatory influence of co-axiality on EVC's pRF anisotropy successfully relates to that in the corresponding co-axial bias in the perceptual anisotropy.

## Discussion

To better understand EVC's contribution to spatial extent perception, we concurrently probed the pRF anisotropy of human individuals and the anisotropy of their spatial extent perception for two factors: radiality and co-axiality. We found that the pRF anisotropy was biased by radiality and weakly influenced by co-axiality, while the perceptual anisotropy was biased by co-axiality and moderately influenced by radiality. Despite these opposing patterns in anisotropy bias, the inter-individual variabilities in the degree of anisotropy for both factors were significantly shared between EVC and perception. Below, we discussed the implications of our findings about the topographic features of human EVC and its contribution to spatial extent perception.

As detailed in the first Results section, the two pieces of knowledge should be considered to properly characterize the pRF anisotropy: (i) the relationship between individual fMRI voxels and orientation columns and (ii) the correlation between the

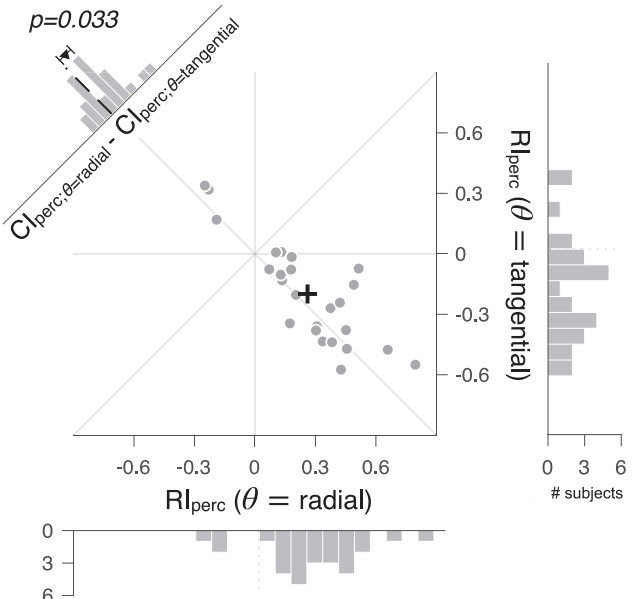

**Fig. 5 Topographic anisotropy in perceived spatial extents.** Across 27 observers, each represented by a single dot, the radial bias indice of perceptual anisotropy in the radial orientation condition, $RI_{perc}(\theta = radial)$, is plotted against that in the tangential orientation condition, $RI_{perc}(\theta = tangential)$. The histograms at the bottom and on the right show the across-individual distributions of $RI_{perc}$ values in the radial and tangential orientation conditions, respectively. Note that most of the $RI_{perc}$ values fall on the positive side in the radial orientation condition and on the negative side in the tangential condition, which indicates that the anisotropy of perceived spatial extents is co-axially biased. Given this co-axial bias, the $RI_{perc}$s were converted into the co-axial bias indices ($CI_{perc}$s) by keeping their signs in the radial orientation condition ($CI_{perc;\theta=radial} = RI_{perc;\theta=radial}$) while flipping them in the tangential condition ($CI_{perc;\theta=tangential} = -1\times RI_{perc;\theta=tangential}$) (read the main text for the rationale). The histogram in the upper-left corner shows the across-individual distribution of the differences in $CI_{perc}$ between the two orientation conditions, $CI_{perc;\theta=radial} - CI_{perc;\theta=tangential}$. Dots, 27 observers; Black crosshairs, across-individual means, and their standard errors.

preferred orientations and polar-angle positions of visual neurons or their connections. In light of this, one must be cautious when interpreting a recent report on the radially elongated pRFs in EVC[22]. Since the pRFs were estimated using the conventional checkerboards in that study, the observation of the radially elongated pRFs per se does not inform about which one of radiality or co-axiality influences EVC's spatial representation. This is because both factors lead to the radially elongated pRF due to the correlation between preferred orientation and polar-angle position (as illustrated in Fig. 1a, b). By contrast, since we estimated the pRFs using the patterns with radial and tangential orientations, our results are free from such indeterminacy (as illustrated in Fig. 1c–f) and thus could conclusively point to radiality as the dominant factor of EVC's pRF anisotropy.

In a related study, the same research group has shown that the degree of radial elongation of pRFs in EVC decreased after EVC was adapted to radially oriented stimuli[21]. The authors interpreted their findings as evidence for the co-axial bias of horizontal connections. However, given that cortical adaptation can alter the sensitivity gain of target neurons as well as the inhibition of their surroundings depending on the spatial configuration of adaptor stimuli[41], the adaptation-based fMRI results invite multiple alternative interpretations. Moreover, the use of full-field grating stimulus in that study has been known to shift the orientation

tunings of V1 neurons toward the adapted orientation[42,43], which is the effect opposite to what the authors intended to create with adaptation. By contrast, by comparing the degree of anisotropy between the radial and tangential orientation conditions (Fig. 3), our study could reveal that co-axiality is not the dominant factor of EVC's pRF anisotropy and exerts only a weak modulatory influence on it (a conclusion corresponding to the case depicted by the right panel of Fig. 1f).

We found that the pRF anisotropy of EVC was biased by radiality while being only weakly influenced by co-axiality. In view of this finding, we gave a thorough re-evaluation of the literature on the co-axial bias in EVC.

Led by a seminal work[26], the anatomical studies on EVC in various species have reported preferential horizontal connections between neurons with co-axially aligned RFs[25,27]. These anatomical findings and ours are reconciled by considering the following aspects. First, the co-axial anisotropy in horizontal connections in primate V1 is either small[27] or negligible[44] compared to non-primate V1[25,26], which suggests that the co-axial anisotropy in horizontal connections may be quite weak or negligible in human EVC. This reconciliation seems plausible given the previously reported inter-species differences in other aspects of functional architecture in EVC, including pinwheel density[45], orientation column organization[46], and excitatory connections[47–49]. Second, co-axial anisotropy in horizontal connections has become a topic of debate[50,51]. For instance, there was no evidence for the co-axial alignment in the retinotopic spatial distribution of pyramidal cells' boutons in cats' EVC[51]. Therefore, caution needs to be exercised when approaching this issue, considering the inter-species differences in the functional architecture of EVC.

Previous neuroimaging studies on the EVC of monkeys[5] and humans[52,53] have demonstrated that EVC's spatial representation of visual input is affected by its orientation texture. As pointed out earlier, determining the dominant factor of anisotropy requires the orthogonal manipulation of stimulus orientation along the main axes defined in the polar space, i.e., radial vs. tangential orientations (Fig. 1c–f). In those studies, although the spatial extents were probed using orientated gratings, the orientation was not manipulated in the polar space. In this regard, what those studies have demonstrated is not the predominance of co-axiality over radiality but the modulatory influence of co-axiality on anisotropy. Thus, their findings do not oppose or contradict our own. For that matter, we also observed this modulatory influence of co-axiality in EVC's pRF anisotropy, although it was quite weak and failed to reach a statistically significant level (as indicated by the histograms in Fig. 3b and the difference between the colored horizontal bars in Fig. 6a). In this sense, our findings are in agreement with theirs, but we have extended them by delineating the distinct natures and degrees of the influences of radiality and co-axiality on the anisotropy in EVC: the former biases it as the dominant factor while the latter exerts only a modulatory influence on it.

In sum, the influences of radiality and co-axiality on EVC's pRF anisotropy observed in our study do not conflict with the results of previous anatomical and functional studies. Instead, our findings suggest that the nature and strength of the co-axial anisotropy in human EVC may need to be re-characterized: its influence on spatial representation may be modulatory in nature and weak in strength.

A pioneering study on the topographic anisotropy of human spatial extent perception[5] rigorously demonstrated that the perceived spatial extent is modulated by co-axiality using a task of circularity judgment, which is adopted in our study (Fig. 4). In that study, the standard and probe gratings had a vertical or horizontal orientation and were presented always on the 45°

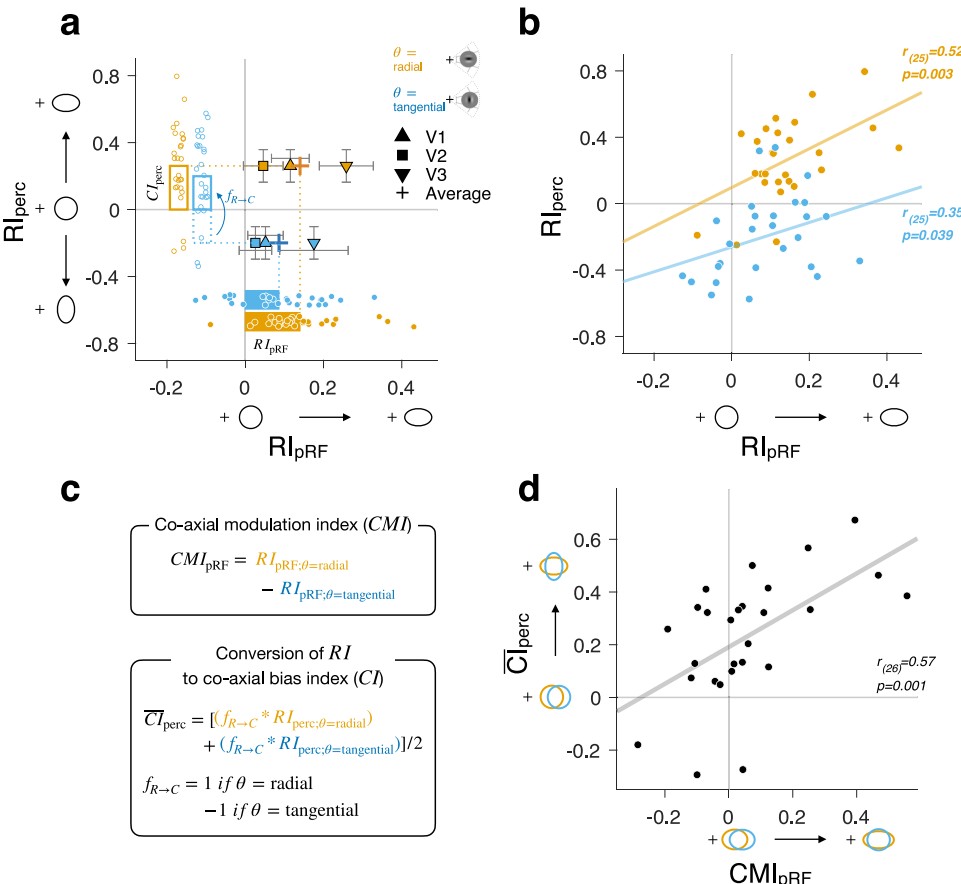

**Fig. 6 A mismatch in bias and co-variations in the inter-individual difference between the pRF anisotropy and the perceptual anisotropy. a** Across-individual averages of the radial bias indices of the pRF ($RI_{pRF}$) and perceptual ($RI_{perc}$) anisotropies in the radial and tangential orientation conditions. The x-axis values of the symbols are the across-individual averages of $RI_{pRF}$ for V1, V2, and V3. As for the y-axis values, the one and same across-individual average of $RI_{perc}$ is repeatedly used for the symbols in each stimulus orientation condition. The solid horizontal bars are the across-individual and across-area averages of $RI_{pRF}$ values for the radial (orange) and tangential (blue) orientation conditions. The empty vertical bars are the across-individual averages of $CI_{perc}$ values for the two orientation conditions. Note that the average value of $CI_{perc}$ in the tangential orientation condition was converted from that of $RI_{perc}$ by flipping the sign (as defined in the bottom panel of **c**). Dots, 27 observers; Symbols and error bars, mean and 95% CI; Upward triangle, square, downward triangle, and crosshair symbols represent V1, V2, V3, and the average of V1-3. **b** Across-individual co-variations between the influences of radiality on the pRF and perceptual anisotropies, shown separately for the radial (orange) and tangential (blue) orientation conditions. For each orientation condition, the $RI_{perc}$ values of 27 individuals are plotted against the averages of their $RI_{pRF}$ values across the visual areas. In both conditions, the $RI_{perc}$ values were significantly regressed onto the $RI_{pRF}$ values, as depicted by the linear regression lines and Pearson correlation values. **c** Definition of the co-axial modulation index of pRF anisotropy ($CMI_{pRF}$) and the mean co-axial bias index of perceptual anisotropy ($\overline{CI}_{perc}$). Top, $CMI_{pRF}$ quantifies the modulatory influence of co-axiality on the pRF anisotropy based on the difference between the $RI_{pRF}$ values of the two orientation conditions. Bottom, $\overline{CI}_{perc}$ quantifies the mean degree of the co-axial bias in the perceptual anisotropy by averaging the $CI_{perc}$ values of the two orientation conditions, which were converted from the corresponding values of $RI_{perc}$. **d** Across-individual co-variations between the influences of co-axiality on the pRF and perceptual anisotropies. The $\overline{CI}_{perc}$ values of 27 individuals are plotted against the averages of their $CMI_{pRF}$ values across the visual areas. The $\overline{CI}_{perc}$ values were significantly regressed onto the $CMI_{pRF}$ values, as depicted by linear regression line and Pearson correlation value. **a, b, d** Schematically depicted ellipsoids are used as guides on the x and y axes to visually illustrate the topographic anisotropies corresponding to the $RI_{pRF}$ and $RI_{perc}$ values (**a, b**) or the $\overline{CI}_{perc}$ and $CMI_{pRF}$ values (**d**).

radial axis in the polar space. In this setup, the gratings with vertical and horizontal orientations appeared elongated vertically and horizontally, respectively. Our study also confirmed that physically isotropic gratings appear elongated along texture orientation based on the data collected from a substantially larger number of individuals (n = 27), using the radially and tangentially oriented gratings presented on the four different radial axes in the polar space (0°, 45°, 90°, and 135°). Thus, our findings support the earlier study[5] by showing that the co-axial anisotropy in spatial extent perception is generalizable and robust.

However, it is worth noting that the current work also goes beyond previous studies by revealing the modulatory influence of radiality on the perceived spatial extent (as indicated by the histograms in Fig. 5 and the difference between the colored

vertical bars in Fig. 6a). We were able to capture this influence because we orthogonally manipulated the orientation of gratings along the radial versus tangential axes in the polar space. The lack of this radial versus tangential orientation manipulation in previous studies[3,5] did not allow for detecting radial anisotropy separately from co-axial anisotropy. This insensitivity to radial anisotropy holds true even when a plaid consisting of vertical and horizontal gratings is presented on the 45° radial axis in the polar space[5]. In that arrangement, the perceived aspect ratio of 1 does not necessarily imply the absence of radial anisotropy because, even in the presence of radial anisotropy, the perceived aspect ratio of the plaid would remain 1 unless there is an imbalance between the co-axial anisotropies in the vertical and horizontal directions.

In sum, our behavioral findings not only strengthen the generalizability and robustness of the earlier demonstration of the co-axial anisotropy in spatial extent perception but also uncover the hidden modulatory influence of radiality on it.

The primary objective of the present work was to refine the existing understanding of how EVC contributes to spatial extent perception by assessing the relationship between EVC and perception in one of the fundamental properties of spatial extent, topographic anisotropy. We evaluated this relationship in two aspects, whether EVC and perception match in the mean bias of anisotropy and whether the inter-individual variability of anisotropy in EVC relates to that in perception. The former evaluation indicates a clear mismatch, the radial bias in EVC and the co-axial bias in perception, while the latter indicates a tight linkage, the co-variations in both radial and co-axial anisotropies between EVC and perception. These two lines of results, when interpreted together with previous work in the context of the hierarchical architecture of visual information processing in the primate visual brain[40], jointly suggest a limited yet critical role of EVC in spatial extent perception.

A large volume of previous work suggested the crucial involvement of EVC in spatial extent perception, based on the exquisite finesse in human spatial extent ratio discrimination commensurate with the high-fidelity space representation in EVC[3,4] and the contextual modulation of the spatial representations of EVC matched to that of perceptual representations[5,7–14]. However, the opposing patterns in anisotropy bias between EVC and perception found in our study clearly bound the contribution of EVC to spatial extent perception. Such bounded roles of EVC in perception have previously been proposed as accounts of otherwise perplexing mismatches between EVC activity and visual perception during binocular rivalry[54], line drawing[55], visual masking[55], orientation estimation[56], and perceptual decision-making[57,58]. We conjecture that there is an additional neural mechanism transforming the radially biased spatial representation into the co-axially biased one, presumably being implemented over the downstream cortical regions beyond EVC along the hierarchy of information processing toward the completion of spatial extent perception. This transformation should involve correcting EVC's spatial representation for the radial bias while augmenting its weak and modulatory co-axial bias. Adjusting the readout weights for spatial extent topographically at the decoding stage may readily help correct for the radial bias, considering the stimulus-independent nature of the radial bias. On the other hand, promoting the co-axial bias may require a more proactive strategy, like selectively sampling neural outputs based on their orientation preference at the decoding stage[40], considering the stimulus-dependent nature of the co-axial bias. Identifying the brain mechanism that can deploy such a strategy will be an exciting future research direction that would advance our understanding of the neural substrate of spatial extent perception.

On a cautionary note, we stress that the current study found the radial bias at the spatial extent of the pRF, which reflects the aggregate receptive field of a large population of neurons in EVC, as well as their horizontal connections. We based EVC's spatial extent representation solely on the anisotropy of pRF, as shown in Fig. 1e, f. The anisotropy of EVC's spatial extent representation may have sources other than the anisotropy of pRF. As one such source, one may consider the cortical magnification factor. However, the cortical magnification factor, in principle, cannot contribute to the anisotropy of EVC's spatial extent representation, as it only affects the mapping function from points in the visual field to those on the cortical surface, which is nullified in its inverse function, i.e., the mapping function from the cortical surface to the visual field.

On the other hand, our finding of the inter-individual variabilities shared between EVC and perception suggests that the spatial representations encoded in EVC reliably propagate to a certain cortical region that forms the spatial representations directly associated with the perceived spatial extents. This successful neural propagation of topographical anisotropy should not be downplayed when considering the information loss through the cascades of additional noises built up along the hierarchy of visual areas[59–61]. In this respect, it is indeed impressive that not only the variability in the predominant radial anisotropy but also that in the weak modulatory co-axial anisotropy relate to the corresponding variabilities in perceptual anisotropy (Fig. 6b, d). As another direction of future work, tracking down the cortical pathway over which these variabilities of topographical anisotropy in EVC propagate would provide important clues about the aforementioned mechanism of radiality-to-co-axiality transformation.

Put together, our findings suggest that the neural signals representing spatial information in EVC propagate over a neural mechanism that transforms the radial bias into the co-axial bias along the hierarchy of the primate visual cortex. It is intriguing to note that an inversed direction of transformation in anisotropy—from the world to the retina—occurs in the course over which image inputs to the brain are generated. The topographical anisotropy of the world images is co-axially biased due to the co-occurring edges of objects[62–64], while that of the retinal images is radially biased due to saccadic eye movements[65]. These mirrored transformations in anisotropy between the image generation (from "co-axial" to "radial") and the neural processing stream (from "radial" to "co-axial") seem to align with the Bayesian-brain view[66–68]. According to this view, the hierarchically organized visual cortex inverts the process of retinal image generation to infer the true properties of objects in the world that produced the retinal images. Under this architectural scheme, the correspondence in anisotropy bias between EVC and the retinal image statistics suggests that the computational role assigned to EVC is not to represent the properties of world objects directly. Instead, its role appears to represent those of the retinal images with high fidelity, forwarding those representations to downstream regions where the properties of world objects are recovered with the knowledge of the process of retinal image generation.

## Methods

**Participants**. Twenty-nine human observers (normal or corrected-to-normal vision; aged 20–30; 15 females) participated in the fMRI experiment. Twenty-seven of them (14 females) also participated in the psychophysical experiment (14 females). All observers except for one (the first author) were naïve to the purpose of the study. They provided informed consent, and the experiments were conducted by the guidelines approved by the Institutional Review Board at Seoul National University.

**MRI acquisition**. MR data were acquired using a 3 T Siemens MAGNETOM Trio at the Seoul National University Brain Imaging Center. For anatomical image scans, a high-resolution T1-w MPRAGE was acquired using a 32-channel head coil ($1 \times 1 \times 1$ mm$^3$ voxel, 1.9 s TR, 2.36 ms TE, 9° flip angle (FA) for 12 observers; $0.85 \times 0.85 \times 0.85$ mm$^3$ voxels, 2.4 s TR, 3.42 ms TE, 8° FA for 17 observers). A 20-channel head coil (only the bottom part of a 32-channel head coil) was used for functional image scans to avoid partial blockage of the visual field. Twenty-four oblique slices orthogonal to the Calcarine sulcus were prescribed at the most posterior part at the occipital pole. GRAPPA, 1.5 s TR, 30 ms TE, 75° FA, $2 \times 2 \times 2$ mm$^3$ voxel size, $96 \times 80$ matrix size, two acceleration factors, and the interleaved slice acquisition

order with 62.5 ms inter-slice interval were used. Functional scan follows in-plane T1-w imaging using a 20-channel head coil (MPRAGE, $1 \times 1 \times 1$ mm$^3$ voxel, 1.6 s TR, 2.36 ms TE, 9° FA). This compensated for head motion between the 32-channel T1-w scans and functional scans.

**Visual stimulus presentation**. Stimuli were generated using MATLAB (MathWorks) in conjunction with PsychToolbox[69] on a Macintosh computer. The stimuli were presented by an LCD projector (Canon XEED SX60; Canon) at its native resolution ($1400 \times 1050$ pixels; refresh rate, 60 Hz) onto a rear projection screen placed inside the magnet bore. The distance to the screen from the eyes was 87 cm, and the projection area on the screen was $34.5$ cm $\times 26$ cm, resulting in a visual angle of 22° (width) $\times 17°$ (height). Observers viewed the stimuli through the front surface of a mirror with a multilayer dielectric reflective coating (Sigma Koki) mounted on the head coil. A custom-made neutral density filter (9% transmission rate; Taeyoung Optics) was inserted between the projector lens and the screen to control the overall level of stimulus luminance. The color lookup table was calibrated to linearize the luminance values at the screen center ranging from $0.0045$ cd m$^{-2}$ to $63.5$ cd m$^{-2}$ using a luminance meter (LS-100; Konica Minolta Sensing in conjunction with in-house software for automated measurement and correction).

**pRF-mapping visual stimulus design**. To assess whether spatial extents of pRF of the V1, V2, and V3 voxels are anisotropic in the polar space, and, if so, how the direction of the strength of such anisotropy depends on stimulus orientation or the polar-angle position of the pRF, we acquired fMRI data from the occipital lobes of 29 human observers while they were viewing radially or tangentially oriented narrowband orientation stimuli drifting slowly around a fixation point. Specifically, instead of counter-phased flickering checkerboards in conventional methods[70], we used multiple orientation carriers that traverse the visual field[52,71]: high-contrast and counter-phased flickering (1/150 ms) Gabor patches drift across the visual field to evoke cortical activities as it crosses over an aggregated RFs of neurons within a given cortical tissue. Since we were interested in the anisotropic spatial extent of pRF over the radial versus tangential axes of the polar space depending on stimulus orientation, we considered two orthogonal orientation conditions defined in terms of polar coordinate system: We used Gabor patches with radial or tangential orientation arranged in the shape of ring or bowtie wedge (the first and the second row in Fig. 2b; Radial orientation condition is shown for example), and ring- or wedge-shaped aperture traverses the visual field in the radial or tangential direction. The aperture follows a periodic pattern and completes a cycle in 24 s, and the 24 s periodic pattern repeats nine times with no interval between cycles. In each cycle, the aperture moved in sync with the fMRI data acquisition (steps in every 1.5 s), and flickering Gabor stimuli in 1/150 ms remained at a given position for 300 ms, and a stimulus-off period of 75 ms was followed (for schematics, see Fig. 2a). The stimuli covered spatially extended region of retinotopic space up to 7.9° in radius.

To provide an effective trigger at any visual field, first, we let a visual stimulus remain at any given visual space for 3 s (2TR). Meanwhile, a ring- or wedge-shaped aperture remained at a given position for 1.5 s (1TR) and was followed by the next aperture in a new position, shifting by the size of the half-width of the aperture. Second, the Gabor size increased linearly as a function of the eccentricity position of the Gabor center, and spatial frequency values were determined by their eccentricity positions by applying the following logarithmic

function: $SF_i = SF_{fovea} - \frac{5}{6}\log(e_i)$. $SF_i$ is the spatial frequency of Gabor located in the eccentricity of $i$, and $SF_{fovea}$ is the spatial frequency at the fovea as 3 cycles per degree (cpd). As a result, the spatial frequency ranged from 1.34 cpd at an eccentricity of 7.3° to 3 cpd at an eccentricity of 1°.

To minimize potential second-order interaction effects between neighboring Gabor patches, e.g., collinear facilitation, we imposed constraints on the spatiotemporal layout of the contrast envelopes of Gabor stimuli, and we randomly varied the phase of Gabor stimulus. Details for ring and wedge scans are provided below, respectively.

For the expanding/contracting ring scan, 16 ring apertures of different sizes centered at eccentricity positions, [1°, 1.30°, 1.63°, 1.95°, 2.31°, 2.67°, 3.05°, 3.44°, 3.86°, 4.29°, 4.75°, 5.21°, 5.71°, 6.21°, 6.76°, 7.30°]. Minor radius of torus-shaped ring stimulus corresponds to [0.3°, 0.31°, 0.33°, 0.34°, 0.36°, 0.37°, 0.39°, 0.40°, 0.42°, 0.44°, 0.46°, 0.48°, 0.50°, 0.52°, 0.55°, 0.57°]. It determines the size of Gabor as two times the standard deviation ($\sigma$) of a 2D Gaussian. The colored circle in Fig. 2a is the outline of a 2D Gaussian envelope containing 95% of the Gaussian filter ($2\sigma$ in radius). Each ring aperture remained at a given eccentricity for 1.5 s (1TR) and expanded or contracted in discrete steps with the same size of Gabor radius, so each ring-shaped stimulus remained at any given visual space for 3 s (2TR) except for the innermost ($<1°$) and outermost ($>7.3°$) eccentricity. At any given eccentricity, four sets of a pack of Gabor stimuli formed a ring aperture (the upper panel in Fig. 2a). To minimize the spatial interaction effects between simultaneously appearing Gabor stimuli, Gabor stimuli in each set were spatially separated more than three times of the Gabor size in radius.

For the clockwise/counter-clockwise rotating wedge scan, the wedge-shaped aperture consisted of two sets of eccentricities (lower panel in Fig. 2a; red/magenta and blue/green). The first set consisted of 7 eccentricities (1°, 1.65°, 2.39°, 3.24°, 4.22°, 5.35°, 6.64°; red/magenta colored), and the second set consists of 6 eccentricities (1.30°, 1.99°, 2.78°, 3.70°, 4.74°, 5.95°; blue/green colored). The size of the Gabor stimulus is determined by [0.30, 0.35, 0.40, 0.46, 0.52, 0.60, 0.69] or [0.32, 0.37, 0.42, 0.49, 0.56, 0.65], which corresponds to two times the standard deviation ($\sigma$) of a 2D Gaussian. The wedge aperture rotated in the discrete step of 11.25°. Only one Gabor stimulus was presented at a given eccentricity, and neighboring Gabor stimuli were arranged in a zigzag pattern to minimize the spatial interaction effects.

The functional MRI experiment consisted of nine scans: eight traveling scans with ring- or wedge-shaped apertures and one hemodynamic impulse response (HIRF) estimation scan. Eight traveling scans included two stimulus orientation conditions, radial and tangential orientation, and four traveling aperture conditions, CW, CCW, EXP, and CONT (CW: clockwise wedge; CCW: counter-clockwise wedge; EXP: expanding ring; CONT: contracting ring). Experiments were conducted in the following fixed order: HIRF, CW_c, CW_r, EXP_r, EXP_c, CCW_c, CCW_r, CONT_r, CONT_c (, where '_r' and '_c' indicate the radial and tangential orientation conditions, respectively). The Gabor orientation was maintained during the nine cycles of each scan to obtain a robust measure of orientation-dependent fMRI time series. The fixation behavior during the experiment was assured by monitoring observers' performance on a fixation task, in which they had to detect any reversal in the direction of one pair of small green dots (0.07° in diameter) on a stationary red tangential circle (0.14° in diameter) at the center of the screen.

We stress that only the stimulus orientation content was manipulated in radial or tangential orientation within a fixed spatiotemporal layout of a Gaussian envelope (e.g., CW_r vs.

CW_c scan). Thus orientation-dependent changes in pRF anisotropy at the same cortical site are due to the differences in neural contribution to the pRF, not due to those in other nuisance factors (e.g., hemodynamic response properties) because they were constant between the orientation conditions.

**MRI data preprocessing.** All functional EPI images were slice-timing corrected and motion-corrected using SPM8[72,73]: The measurement times for individual images were corrected by shifting the phase of all frequency components, and then 3D rigid-body transformations were performed to realign the image frame to the first frame of the scan. After correction, the functional data from all scans aligned to the T1-w anatomical image by application of warp matrix from the high-resolution in-plane image to the reference image[74]. In-house analysis code was used in conjunction with mrTools analysis package (http://gru.stanford.edu/doku.php/mrtools). The cortical surface was constructed from a 1mm³ or 0.85mm³ T1-w image rendering of each observer using Freesurfer[75,76] to create a flattened representation of the occipital lobe. Cortical segmentation of the inner and outer layers of the gray matter was done using Jonas Larsson's SurfRelax. This information was used in mrLoadRet to display a flattened representation of the occipital cortex.

We manually defined the boundaries between V1, V2, and V3 on the flattened gray matter cortical surface based on the meridian representations by analyzing the temporal phases of fMRI responses to rotating wedge stimuli[77].

**Voxel selection for pRF analysis.** The first 16 frames of each scan were discarded due to initial magnetization transients. Individual voxels' fMRI time series were converted into percent signal changes and linearly detrended to correct the slow non-physiological baseline activity components.

We applied the following two criteria when selecting valid voxels for further analysis. First, the voxels that were substantially modulated with the cyclic visual stimuli were selected: (i) the correlation between their measured time series and the best-fitting sinusoidal function should be greater than 0.4; (ii) the temporal phase of their HIRF should be within $\pm \pi/4$ around the phase of the stimulus onset; (iii) their signal-to-noise ratio (SNR) should be greater than 2, where the SNR was computed in a Fourier domain by dividing the amplitude of the stimulus frequency component (1/24 s) with the average amplitude of the frequency components three times higher than the stimulus frequency for each voxel. Second, the voxels that include large draining veins near the pial surface were discarded from the analysis because their fMRI signals are decoupled from neural activity both in the spatial[78,79] and temporal domains[80]: the voxels were classified as blood-vessel-clamping if their variance was in the highest 10% out of the entire voxels. As a result, 55.2% ± 6.3% of total voxels (across individuals, mean ± s.d.) were used for further analysis.

**Observer-specific HIRF estimation.** For HIRF estimation, 144 frames of images were acquired before the retinotopy mapping scan, and the first cycle (16 frames) was discarded due to initial magnetization transients. The hemodynamic response was driven by a briefly pulsed (3 s in duration, two times the fMRI sampling rate) full-field Gabor patches. The stimulus pulse was repeated nine times, separated by 21-s intervals. We modeled HIRF as a difference of two gamma functions with six free parameters[81,82] and fitted the predicted fMRI time series to the observed time series using the least-square procedure (MATLAB's fminsearch.m). Since HIRF can vary substantially across observers[83,84] and the pRF estimation (especially spatial extents of pRF) is

affected by HIRF[85,86], we estimated HIRFs separately for individual observers. The percent variance explained by the estimated HIRFs was 97.68 ± 0.01% (mean ± s.d. across 29 observers).

**Estimation of pRF.** We conducted a model-based pRF analysis that was developed to estimate the stimulus-referred aggregate RF of a population of neurons within a voxel[70] in the following procedure. Initially, we fitted an isotropic 2D Gaussian model to the fMRI time series to determine the center position of pRF. Next, we fitted an elliptical 2D Gaussian model to the times series with the center position defined in the initial step to estimate the anisotropy of pRF in the polar space.

At the initial stage of pRF estimation, the isotropic 2D Gaussian model was defined by two parameters associated with the center position of pRF, $x_0$ and $y_0$, and another with the width of pRF, $\sigma$, as follows: $g(x,y) = e^{-\frac{(x-x_0)^2+(y-y_0)^2}{2\sigma^2}}$. A linear overlap between the pRF, $g(x,y)$, and a binary mask, $s(x,y,t)$, of stimuli across time predicts the response of neural population at each voxel, where $s(x,y,t)$ represents the spatial locations of Gabor patches, assigning 1 and 0 to the points inside and outside, respectively, its Gaussian envelope ($2\sigma$ in radius). Thus, the linear overlap between $g(x,y)$ and $s(x,y,t)$ predicts the time series of aggregated neurons' responses within a given voxel. The temporal convolution of this time series with the observer-specific HIRF, $h(t)$, predicts the time series of fMRI response, $y(t)$: $y(t) = h(t) * \sum_{x,y} s(x,y,t)g(x,y)$, where $*$ is convolution operation. We found the set of best-fitting parameters using one minus correlation coefficient between observed and predicted BOLD responses as a cost function. We used MATLAB's fminsearchbnd.m function to constrain the range of parameter values with our prior knowledge. As for the center position of pRF, the eccentricity was constrained to [0.7°, 7.9°], given the visual field coverage of stimuli (see Visual stimulus design in Materials and Methods). As for the width of pRF, $\sigma$ was constrained to [0.1°, 5°].

At the second stage of pRF estimation, the elliptical 2D Gaussian model was defined by the two central position parameters of pRF, $x_0$ and $y_0$, whose values were inherited from the previous stage of fitting, and the two free parameters, $\sigma$ and $AR$ (aspect ratio). As we were interested in the anisotropy of pRF along the radial versus tangential axes, the elongation axis was constrained to one of the two axes. The aspect ratio of pRF was defined as $\sigma_{\text{radial−axis}}/\sigma_{\text{tangential−axis}}$, where $\sigma_{\text{radial−axis}}$ and $\sigma_{\text{tangential−axis}}$ were the radial and tangential spatial extents of pRF, respectively. To elongate pRF radially or tangentially, we first took new x and y coordinates after counter-clockwise rotation angular position ($\theta$), by applying $2 \times 2$ rotation matrix, $\mathbf{R} = \begin{bmatrix} \cos(\theta) & -\sin(\theta) \\ \sin(\theta) & \cos(\theta) \end{bmatrix} : \begin{bmatrix} x' \\ y' \end{bmatrix} = \mathbf{R}\begin{bmatrix} x \\ y \end{bmatrix}$. In the same manner, new coordinates of pRF position were obtained as $\begin{bmatrix} x_0' \\ y_0' \end{bmatrix} = \mathbf{R}\begin{bmatrix} x_0 \\ y_0 \end{bmatrix}$. As a result, the elliptical 2D pRF model was defined as $g(x',y') = e^{-\frac{(x'-x_0')^2}{2(\sigma\sqrt{AR})^2} - \frac{(y'-y_0')^2}{2(\sigma/\sqrt{AR})^2}}$. We constrained the absolute log aspect ratio to be smaller than 2, based on the previous findings where it ranged between 1–1.67 for EVC[21,22,87]. However, our results were not affected when the range was extended up to 3.

We selected valid voxels for further analysis only when the correlation coefficient between observed and predicted data at a given voxel was greater than 0.25 in all functional scans. Our results were also confirmed using a more conservative criterion ($r > 0.3$). As a result, 28.1% ± 7.6% of the entire voxels in EVC (across-observer mean and s.d.; 495.4 ± 157.1 in voxel number)

were included for pRF analysis (see Voxel selection for pRF analysis in Materials and Methods).

**Calculation of the pRF anisotropy based on the subtractive contrast between the radial and tangential spatial extents of pRF.** While previous fMRI studies quantified the pRF anisotropy with divisive contrast[21,22,87,88], we quantified it with subtractive contrast. Unlike these studies, the primary object of the present work is to evaluate the contribution of the neural representation of space in EVC to the perceptual representation of spatial extent based on the correspondence in topographic anisotropy. Under this aim, the pRF anisotropy expressed in subtractive contrast, compared to that expressed in divisive contrast, better captures the property of the neural representation that is relevant to the topographical anisotropy in spatial extent perception. This is because the visual neurons that contribute most to the topographical anisotropy of neural responses to a visual stimulus are those whose RFs are positioned around the edge of the stimulus, and thus the influence of those edge-representing neurons on the anisotropy of neural responses is proportional to the difference, rather than the ratio, between the pRF widths in two orthogonal directions. Furthermore, the anisotropy measure based on divisive contrast has a scaling issue: it is susceptible to pRF size, being increasingly underestimated as pRF size increases.

**Stimuli for psychophysical experiment.** Stimuli were gray-scale images presented on a calibrated LCD monitor located 70 cm from observers and set to a resolution of $1280 \times 1024$ pixels (pixel size $= 0.295 \times 0.293$ mm). Each stimulus display consisted of a pair of targets, each positioned 3.5° from fixation and centered oppositely. The target stimuli have the same orientation of radial or tangential with a spatial frequency of 1.96 cpd, which were windowed by a two-dimensional Gaussian contrast function whose s.d. in the radial and tangential axes had a geometric average of 0.2 degrees. All stimuli were presented at 80% contrast against a uniform gray background with a mean luminance of 40 cd m$^{-2}$. All stimulus properties were matched with fMRI experimental design.

**Task.** Observers participated in a circularity-discrimination task (Fig. 4a; adapted from Michel et al.[5]). On each trial of the task, while fixating on the display center, the observer viewed a pair of Gabor stimuli briefly (200 ms) presented at mirrored locations on the radial axis with respect to the fixation and judged which one has a shape closer to a circle. The paired stimuli always had the same orientation, either radial or tangential, but can have different contrast envelopes: one with an isotropic 2D Gaussian envelope (standard) and the other with a radially or tangentially elongated ellipsoidal envelope (probe). The probe's radial-to-tangential aspect ratio (AR) was selected randomly from 9 values, which are equally spaced between 0.5 and 2 on a logarithmic scale (0.5, 0.59, 0.70, 0.84, 1, 1.19, 1.41, 1.68, 2).

All observers first performed a practice experiment (50 trials per block, up to 8 blocks) with correct, incorrect, or late feedback. In case of an incorrect response, the observer can learn the task with an explanation for the incorrect feedback. The observer was allowed to start the main experiment only if the task performance of the observer met the criterion with 75% correct or more. In the main experiment (100 trials per block, a total of 10 blocks), there was no feedback for each trial, and only overall performance in each block was presented. This design is to minimize the effect of understanding instruction. Participants reported which one has a shape closer to a circle by pressing the left or right arrow key when a pair of Gabor stimuli presented in horizontal or oblique meridians. When the stimuli were presented in vertical meridian, upward and downward arrow keys were used to receive their report.

**Analysis of behavioral data: perceived anisotropy estimation by fitting psychometric curve.** For each observer, we estimated the AR of the probe that appears to be a perfect circle. To compute the fractions of choosing the standard as a more circular one as a function of the AR of the probe (Fig. 4c), we assumed that the decision is based on noisy perceptual estimates of the AR[5]. The noise ($\sigma$) was proportional to the absolute value of the logarithm of perceived AR and distributed log normally. Thus, perceptual noise is the least about the Gabor stimulus with a perceived AR of 1, given that aspect ratio discrimination is best at $AR = 1$[3]. That is,

$$\log\left(AR_{\text{probe}}\right) = X_P \sim N\left(\mu_p + \log_2(\beta), \sigma\right),$$

$$\log\left(AR_{\text{standard}}\right) = X_S \sim N\left(0 + \log_2(\beta), \sigma\right),$$

where $\mu_p = \log(AR_{\text{probe}})$, and $\beta$ is the perceived aspect ratio of the standard stimulus.

The observer's task was to choose a stimulus with a more circular envelope. It is equivalent to determining which of the two log aspect ratios is closer to zero. In other words, the observer selects the standard whenever $X_S^2 > X_P^2$ and chooses the probe otherwise. We can compute the probability of choosing the standard as below:

$$p(X_S^2 < X_P^2 | \mu_p, \beta, \sigma) = \int_{-\infty}^{\infty} p(X_S^2 < x_p^2 | \beta, \sigma) p(x_p^2 | \mu_p, \beta, \sigma) dx_p.$$

We estimated three free parameter values, [$\sigma$, $\beta$, const.] to maximize the sum of log-likelihoods of the observed data for each individual observer and each stimulus orientation condition using MATLAB's fminsearch.m.

$$p(\text{standard} | \mu_p, \beta, \sigma) = p(X_S^2 < X_P^2 | \mu_p, \beta, \sigma) + \text{const}.$$

**Reporting summary.** Further information on research design is available in the Nature Portfolio Reporting Summary linked to this article.

## Data availability

Our preprocessed retinotopy and psychophysical data are publicly available on the Open Science Framework (https://osf.io/wrdt9/[89]). After request, raw data can be made available with a data-sharing agreement with the corresponding author. Numerical source data underlying the graphs in the manuscript can be found in Supplementary Data File 1.

## Code availability

The open-source packages used in our analysis are stated and cited in Materials and Methods. Custom MATLAB scripts for analyzing the data are publicly available on the Open Science Framework (https://osf.io/wrdt9/[89]).

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

## Acknowledgements
This work was supported by the National Research Foundation of Korea (NRF) funded by the Ministry of Science and ICT (No. NRF-2015M3C7A1031969, NRF-2017M3C7A1047860, NRF-2021R1F1A1052020), and SNU R&DB Foundation (339-20220013).

## Author contributions
J.R. and S.H.L. designed research; J.R. and S.H.L. performed research; J.R. analyzed data and wrote the first draft of the paper; J.R. and S.H.L. wrote the paper.

## Competing interests
The authors declare no competing interests.
