## [Peer review file · Communications Biology]

Reviewers' comments:

Reviewer #1 (Remarks to the Author):

This paper examines the neural substrates of spatial extent perception, testing whether spatial representations in early visual areas (V1, V2, V3) exhibit primarily a coaxial or a radial bias. Based on a pRF analysis, the results demonstrate a radial bias in early visual cortical representations, while a behavioral experiment revealed that spatial extent perception was largely determined by a coaxial bias. However, the results show that the perceived radial elongation of peripheral stimuli was correlated with the elongation of pRFs in early visual cortex. Overall, I think the paper is interesting, clear, and represents a valuable contribution to the literature. In particular, I think the correlation between the anisotropy in pRFs and perceived aspect ratio in the behavioral task is compelling. However, I have some suggestions for strengthening the framing and presentation of the results.

Introduction: The manuscript indicates in multiple places (lines 532, 575, 647), that the imaging results are fundamentally inconsistent with the EVC hypothesis, but the description of the EVC hypothesis seems to take a somewhat narrow interpretation of previous work. I think it would help here to unpack what is meant by the EVC hypothesis and the evidence to support it. At least some of the work cited in support of the EVC hypothesis seems to argue that spatial extent perception correlates with representations in early visual areas, or that activity in these areas contributes to spatial extent perception, which is similar to the results shown here. To that extent, the results appear to be very consistent with previous studies. In addition, the results and discussion sections provide a thorough explanation of the processes underlying spatial extent perception (lines 540-549; 655-662), and some of this could be worked into the introduction of the paper to motivate the experiments more clearly.

Discussion: with this in mind, I think the introduction and discussion could be streamlined to make the interpretation of the results clearer for readers, as it seems to go a bit back and forth about how the results relate to previous findings (i.e., which results are consistent or inconsistent with previous work). Again, I think being a bit clearer about the evidence in favor of the EVC hypothesis would help here. Some other elements of the discussion could also be rewritten for clarity (e.g., the section on the distinction between 'co-axially elongated spatial extents' and 'co-axial modulation of spatial extents', as well as the last paragraph on "the correspondence between 'the image statistics in the world versus retina' and 'the spatial biases in perception versus EVC'").

Methods/Results: While the level of detail is helpful, I think several elements could be moved entirely into the supplemental section and only briefly mentioned in the manuscript in a sentence or two (lines 256-265; 462-475)

More comments:

Figure 6 – it would be helpful to include linear fits to these graphs

Figure 6 caption, line 555: the word "observer" appears twice

Reviewer #2 (Remarks to the Author):

This manuscript describes an elegant study that combines fMRI and psychophysics in human subjects to examine the nature of topographic representations of the visual field in early visual cortex (EVC) and the role of these representations in the perception of spatial extent. The study is carefully designed and executed, the analyses are appropriate and the results are interesting. The key finding is that there is a mismatch in the effects of two known biases that govern topographic representations in EVC – the radial bias dominates physiologically while the co-axial bias dominates perceptually.

While the paper is of significant interest, there are several areas in which it can be improved. First,

some of the language regarding the conclusions is confusing and somewhat contradictory. On one hand, the authors claim that their results are inconsistent with the hypothesis that the spread of activity in the topographic maps in EVC contributes to spatial extent perception ("the EVC hypothesis"). But then in the discussion, they acknowledge that "our findings should not be taken as evidence for the lack of EVC's contribution to spatial extent perception". I believe that this latter conclusion is more appropriate and that the claim that their results are inconsistent with the EVC hypothesis should be toned down and qualified (see additional details in Specific Comments below). Similarly, in some places, the impression is that the authors claim that the co-axial bias does not exist physiologically in EVC, while in others the claim is that it exists but is weaker than the radial bias. The language should be corrected so that it is consistent throughout the manuscript.

Second, the finding that perceptual judgement of spatial extent is affected by the co-axial bias has already been described in a previous study which the current study builds on (Michel et al, 2013). This should be mentioned more clearly in the manuscript. Michel et al, 2013 also contains evidence against a radial perceptual bias, as the aspect ratio of non-oriented visual stimuli was judged veridically by human subjects despite the radial bias in EVC representations. Similarly, other studies that showed high-fidelity perception of circularity (Regan et al, 1992, 1996) already suggested that the physiological radial bias is discounted perceptually. These previous results should be properly mentioned and the way in which the current study expands on these results should be discussed.

Third, some of the technical details are difficult to follow. For example, the analysis of the across-observer correlation in Fig. 6 is not explained clearly. What exactly is being correlated and what does this result mean? Similarly, the two analyses mentioned in lines 649-654 in the discussion are not clearly explained, motivated and interpreted. These analyses seem to be central to the conclusion of the paper and clarifying these issues will help in assessing the significance and novelty of the findings.

Specific comments

- In lines 480-481, the authors write: "perceived spatial extent must be subject to the topographic bias in neural properties". I think that this statement needs to be qualified. It is important for the authors to distinguish between static anisotropies, such as the radial bias, and stimulus-dependent anisotropies, such as the co-axial bias. If the EVC hypothesis is true, decoding circuits that receive information from EVC must take into account such static anisotropies. The naïve hypothesis that there would be a simple mapping between spread of activity in EVC and spatial extent perception that ignores such static anisotropies and distortions is not tenable and can easily be rejected. A more realistic test of the EVC hypothesis is whether stimulus-specific physiological biases such as the axial bias in EVC correlate with perception, and the results of the current study, which confirm the findings of Michel et al 2013, demonstrate that such perceptual biases exist and are robust, providing support to the EVC hypothesis. As discussed above, these issues need to be explained more clearly throughout the manuscript.
- While the co-axial bias effects in EVC are weak and may not reach statistical significance in the current study which uses fMRI, it is important to mention that these effects are robust in macaque V1 when measured with optical imaging (Michele et al, 2013).
- Lines 642-643: "co-axial bias may exist but is not as strong as previously thought". The authors should need to be more explicit and explain which expectations they are referring to.
- Was the analysis of the pRF restricted to a single eccentricity as in Fig. 3A? If so, why? If not, how were the results combined across eccentricities given that the size of the pRF changes with eccentricity?

Dear Editors and Reviewers:

We would like to express our gratitude to the reviewers for their thoughtful feedback. Their comments were constructive and helpful, allowing us to present our work in a more rigorous and convincing way. We have made significant revisions to our manuscript in response to the reviewers' comments, as reflected in the new title. Upon reviewing the concerns raised by the reviewers, we realized that most of them related to the narrowly defined EVC hypothesis that we had used to lead the story of our work. To address this, we have reframed our work by withdrawing the EVC hypothesis and made substantial changes to the manuscript to improve its overall coherence. To give a brief overview, the major changes in the revised manuscript are as follows.

In the Introduction section, we have clarified that our objective is not to test the narrowly defined EVC hypothesis but rather to refine the current understanding of EVC's contribution to spatial extent perception by concurrently inspecting the topographic anisotropy of EVC and perception. We have then introduced two crucial matters that need to be resolved to achieve this objective and explained our approach to address them and assess EVC's contribution to spatial extent perception.

In the Results section, we have made major revisions to two aspects. Firstly, when assessing the influences of the two anisotropy factors (radiality and co-axiality) on the spatial representations of EVC and perception, we've clearly distinguished between the 'biasing' and 'modulatory' influences (as illustrated in the revised Fig. 1e,f and reflected in the revised Results subsection titles). Second, when examining the relationship in inter-individual variability between the EVC's and perceptual anisotropy, we separately assessed the contributions of EVC to perception for the two factors by quantifying the degrees of radial and co-axial influences with corresponding indices (as reflected in the revised Fig. 6). These two changes improved the manuscript by providing more convincing evidence to support our main claims.

We have significantly revised all subsections of the Discussion, to reflect the redefined aims in the Introduction section and the refined outcomes in the Results section. We want to draw attention to the following three points of revision. Firstly, we have clarified the shared and differing aspects between our findings and previous research, and explained how we expanded on the previous work. Secondly, we have summarized our results into two equally weighted pieces of findings: (i) the mismatch in anisotropy bias between EVC and perception (as depicted in Fig. 6a) and (ii) the shared variability in the degree of anisotropy between EVC and perception (as depicted in Fig. 6b,d). Thirdly, we have derived our integrative interpretation from (i) and (ii) and concluded that while the contribution of EVC to perceptual spatial extent is critical in (i), it is clearly bounded in (ii).

Our responses to each reviewer's comments are detailed below, along with explanations of how we revised the manuscript to address them. To specify and cross-reference the reviewers' comments, we have labeled them in a specific format, [R#-C#], where #s correspond to the reviewer's ID and the order of their comment. If a comment needs to be further subdivided, we add "-#" to the end of the label.

Reviewer #1 (Remarks to the Author):

[General comments] "This paper examines the neural substrates of spatial extent perception, testing whether spatial representations in early visual areas (V1, V2, V3) exhibit primarily a coaxial or a radial bias. Based on a

pRF analysis, the results demonstrate a radial bias in early visual cortical representations, while a behavioral experiment revealed that spatial extent perception was largely determined by a coaxial bias. However, the results show that the perceived radial elongation of peripheral stimuli was correlated with the elongation of pRFs in early visual cortex. Overall, I think the paper is interesting, clear, and represents a valuable contribution to the literature. In particular, I think the correlation between the anisotropy in pRFs and perceived aspect ratio in the behavioral task is compelling. However, I have some suggestions for strengthening the framing and presentation of the results.”

Our reply to [General comments] We appreciate the reviewer's constructive feedback and have revised the manuscript accordingly. Below, we detail the changes made to address each suggestion.

[R1-C1] “Introduction: The manuscript indicates in multiple places (lines 532, 575, 647), that the imaging results are fundamentally inconsistent with the EVC hypothesis, but the description of the EVC hypothesis seems to take a somewhat narrow interpretation of previous work. I think it would help here to unpack what is meant by the EVC hypothesis and the evidence to support it. At least some of the work cited in support of the EVC hypothesis seems to argue that spatial extent perception correlates with representations in early visual areas, or that activity in these areas contributes to spatial extent perception, which is similar to the results shown here. To that extent, the results appear to be very consistent with previous studies. In addition, the results and discussion sections provide a thorough explanation of the processes underlying spatial extent perception (lines 540-549; 655-662), and some of this could be worked into the introduction of the paper to motivate the experiments more clearly.”

Our reply to [R1-C1] We acknowledge that we framed our findings in the context of examining the EVC hypothesis and that we had to narrow down our definition of the EVC hypothesis to lend weight to our claims within that framing. After consideration of the reviewer’s comment, we now recognize that our narrow definition of the EVC hypothesis caused us to downplay the aspects of our findings that are consistent with previous studies. We also recognize that our framing strategy caused some confusion in readers’ minds as we went “a bit back and forth about how the results relate to previous findings” throughout the manuscript, as pointed out by the reviewer in the following comment (**[R1-C2]**). As a result, even though the later parts of the Results and Discussion sections accurately represent our ultimate claim based on our results, our framing strategy has made it harder for readers to understand our claim.

Having recognized these multiple problems caused by the framing strategy, we’ve come up with a new approach to transparently deliver our main claim without provoking those problems. At its core, our new approach involves withdrawing the EVC hypothesis. Reframing the manuscript with the new approach described above, we made substantial revisions to the Introduction section (and Fig. 1), as follows.

First, we simply introduced the contributions of EVC to spatial extent perception identified by previous studies (lines 28-34). We then mentioned the implications of such EVC’s roles suggested by those studies in the context of topographic anisotropy (lines 79-81) and pointed out that those implications have been mostly unexplored to justify the need for our work (lines 118-174). Under this new approach, our goal is not to test the narrowly defined EVC hypothesis anymore. Rather, our goal is to refine the contribution of EVC to spatial extent perception by inspecting the topographic anisotropy of EVC and perception (lines 93-99). We then introduced two key issues that need to be addressed to achieve that goal. The first issue pertains to the proper experimental design required to assess the respective influences of the two anisotropy factors, namely ‘radiality’

and ‘co-axiality,’ on EVC’s spatial representation (lines 100-117). The second issue concerns the importance of concurrently inspecting the anisotropy in both EVC and perception from the same individuals (lines 118-174). Lastly, we described how we addressed those two issues and how we plan to assess the contribution of EVC to spatial extent perception (lines 174-183).

Through these revisions, we believe that we’ve made the Introduction more transparent and effective in setting the stage for the rest of the manuscript while addressing the reviewer’s concerns. Specifically, the revisions have helped in (i) avoiding any possible confusion about our claim or misinterpretations of our findings due to the narrow definition of the EVC hypothesis; (ii) highlighting the consistency between previous work and our work; (iii) presenting a more streamlined story flow from the Introduction section to the Discussion section.

[R1-C2] “Discussion: with this in mind, I think the introduction and discussion could be streamlined to make the interpretation of the results clearer for readers, as it seems to go a bit back and forth about how the results relate to previous findings (i.e., which results are consistent or inconsistent with previous work). Again, I think being a bit clearer about the evidence in favor of the EVC hypothesis would help here.”

Our reply to [R1-C2] This comment is related to the comment on the Introduction section ([R1-C1]). As we mentioned in our reply to [R1-C1], we re-defined the goal of our study, without framing it with the EVC hypothesis anymore. With this re-defined goal, which is to refine the current understanding of EVC’s contribution to spatial extent perception, we presented our results into two pieces of findings, with equal emphasis on each. To sum, our findings are: (i) the EVC and perceptual anisotropies are influenced by both anisotropy factors but differ in bias, and (ii) they share the inter-individual variability in the degree of those influences (as shown in the revised Fig. 6). Accordingly, we discussed previous work in this context in the Discussion section, as follows.

Firstly, we clearly highlighted the aspects in which our findings are consistent with previous neuroimaging studies (Michel et al., 2013) and humans (Park et al., 2013; Dumoulin et al., 2014). We have then specified how our study has extended their work by refining the nature of topographic anisotropy in EVC (as stated in the third paragraph of the Discussion subsection titled “**A fair share of the co-axial anisotropy in the spatial representation of EVC**”; lines 1012-1078). Secondly, we also emphasized the consistency between our research and the study conducted by Michel et al. (2013) and how our findings have extended their work by refining the nature of topographic anisotropy in spatial extent perception (as stated in the Discussion subsection titled “**Topographic anisotropy in human spatial extent perception**”; lines 1084-1110). Lastly, we presented the crucial involvement of EVC in spatial extent perception as a fact established by previous studies. We have then pointed out in what way our findings further refined the nature of EVC’s contribution to spatial extent perception (as stated in the first two paragraphs of the Discussion subsection titled “**Bounded contribution of EVC to spatial extent perception**”; lines 1112-1173).

Through these revisions, we believe that the Discussion section now better emphasizes the similarities and differences with previous studies, while clearly presenting our findings and claim in the context set up in the Introduction section.

[R1-C3] “Some other elements of the discussion could also be rewritten for clarity (e.g., the section on the distinction between ‘co-axially elongated spatial extents’ and ‘co-axial modulation of spatial extents’, as

well as the last paragraph on “the correspondence between ‘the image statistics in the world versus retina’ and ‘the spatial biases in perception versus EVC’”).”

Our reply to [R1-C3] We concur with the reviewer that we should have provided our views more clearly by elaborating on the concepts that we used to lead our discussion. We revised those two parts as follows.

Regarding the first part, by using two concepts, namely, ‘co-axially elongated spatial extents’ and ‘co-axial modulation of spatial extents,’ we wanted to convey that previous studies (Michel et al., 2013; Park et al., 2013; Dumoulin et al., 2014) did not demonstrate that ‘co-axiality’ determined the directional bias of the EVC anisotropy, but rather that ‘co-axiality’ had a modulatory influence on the EVC anisotropy. However, upon reflection, we realized that the distinction between those two concepts was not clear enough to deliver our intended messages. This realization highlighted the need for a more explicit differentiation between the ‘biasing’ and ‘modulatory’ influences of the two anisotropy factors on the pRF anisotropy in EVC. To address this issue, we decided to clearly illustrate our definition of their influences from the beginning of the manuscript, as visualized in the revised Fig. 1e,f. We further elaborated on this taxonomy of anisotropy influences in the first Results subsection, titled “**The rationale for pRF estimation using radially and tangentially oriented gratings**” (lines 261-271). Throughout the rest of the manuscript, we described our results by indicating which factors are the ‘biasing’ and ‘modulatory’ factors, as reflected in the revised Results subsection titles such as “**The pRF anisotropy of EVC is radially biased, with a weak modulation by co-axiality**” (line 273) and “**The anisotropy of perceived spatial extent is co-axially biased, with a moderate modulation by radially**” (lines 584-585). In the same vein, we substantially revised the third paragraph of the Discussion subsection titled “**A fair share of the co-axial anisotropy in the spatial representation of EVC**” (lines 1012-1078) to highlight the shared and differing aspects between our findings and previous work using our more explicitly defined terms (as mentioned earlier in our reply to [R1-C2]), instead of using the confusing terms, namely, ‘co-axially elongated spatial extents’ and ‘co-axial modulation of spatial extents,’ in the previous manuscript.

As for the second part, we acknowledge that the link between the world and retinal image statistics and the EVC and perceptual anisotropy biases was depicted too tersely for readers to readily comprehend why and how our findings can be considered consistent with the Bayesian-brain perspective. Initially, we intended to be concise in this part since our opinion is mostly speculative. However, to hit a balance between the need for clarity and brevity, we elaborated more on that link by introducing a concept, namely, “mirrored transformation in anisotropy between the image generation and the neural processing stream” (line 1207). We’ve also provided more specific references to previous studies that support our conceptualizations (line 1191). We believe that our revisions have made it specific enough for readers to understand our somewhat large-scale theoretical take on our findings, while also hoping that the reviewer understands our intention to not elaborate too much on our speculative thoughts.

[R1-C4] “Methods/Results: While the level of detail is helpful, I think several elements could be moved entirely into the supplemental section and only briefly mentioned in the manuscript in a sentence or two (lines 256-265; 462-475).”

Our reply to [R1-C4] We agree with the reviewer’s suggestion that relocating those two parts of the manuscript would improve its flow by removing unnecessary details.

Therefore, we've moved the entire Materials and Methods subsection titled "**Split-half reliability ...**" to the supplemental material (Supplementary Figure 1). We've then briefly mentioned this in the Results section (in the last sentence of the second paragraph of the subsection titled "**The pRF anisotropy of EVC is radially biased, with a weak modulation by co-axiality**"; lines 343-344).

As for the paragraph regarding the proper measure of anisotropy, we've relocated it to the Materials and Methods subsection titled "**Calculation of the pRF anisotropy based on the subtractive contrast ...**" We've then briefly mentioned this in the Results section (the second sentence of the fourth paragraph in the subsection titled "**The pRF anisotropy of EVC is radially biased, with a weak modulation by co-axiality**"; lines 360-362).

As a matter of organization, we've relocated the Materials and Methods section to the end of the Discussion section to improve the flow of the manuscript.

[R1-C4] "More comments:

Figure 6 – it would be helpful to include linear fits to these graphs

Figure 6 caption, line 555: the word "observer" appears twice"

Our reply to [R1-C4] We thank the reviewer for the suggestion and for bringing the typo to our attention. We've added the regression line in Fig. 6 (Fig. 6b,d in the revised manuscript). Additionally, we've also added schematic ellipsoids to help readers better understand the x and y axes. We've corrected the figure caption for that error. We want to emphasize that we've conducted a thorough search for typos throughout the entire manuscript for typos and have made all necessary corrections.

Reviewer #2 (Remarks to the Author):

[General comments] "This manuscript describes an elegant study that combines fMRI and psychophysics in human subjects to examine the nature of topographic representations of the visual field in early visual cortex (EVC) and the role of these representations in the perception of spatial extent. The study is carefully designed and executed, the analyses are appropriate and the results are interesting. The key finding is that there is a mismatch in the effects of two known biases that govern topographic representations in EVC – the radial bias dominates physiologically while the co-axial bias dominates perceptually.

While the paper is of significant interest, there are several areas in which it can be improved."

Our reply to [General comments] We appreciate the reviewer's recognition of our work's merit and constructive suggestions for revision. Below, we address the reviewer's specific comments and describe how we revised the manuscript in response to those comments.

[R2-C1] "First, some of the language regarding the conclusions is confusing and somewhat contradictory. On one hand, the authors claim that their results are inconsistent with the hypothesis that the spread of activity in the

topographic maps in EVC contributes to spatial extent perception (“the EVC hypothesis”). But then in the discussion, they acknowledge that “our findings should not be taken as evidence for the lack of EVC’s contribution to spatial extent perception”. I believe that this latter conclusion is more appropriate and that the claim that their results are inconsistent with the EVC hypothesis should be toned down and qualified (see additional details in Specific Comments below). Similarly, in some places, the impression is that the authors claim that the co-axial bias does not exist physiologically in EVC, while in others the claim is that it exists but is weaker than the radial bias. The language should be corrected so that it is consistent throughout the manuscript.”

Our reply to [R2-C1] In our previously submitted manuscript, we followed a strategy of gradually shaping up our claims against the narrowly defined EVC hypothesis throughout the entire manuscript. We admit that this strategy led to confusion and made our statements appear contradictory, as noted by the reviewer. This comment aligns with the other reviewer’s concerns (see [R1-C1] and [R1-C2]). As described in our replies to [R1-C1] and [R1-C2], we addressed the problems by taking a new approach to clearly present our main claim in a straightforward manner, rather than presenting partial claims piecemeal. Accordingly, we’ve made substantial revisions to the entire manuscript, including the two specific parts highlighted by the reviewer. Below, we detail how we revised each of the two parts.

Regarding EVC’s contribution to spatial extent perception, we explicitly, as a main claim, stated that EVC plays “a limited yet critical role in spatial extent perception.” We arrived at this conclusion by considering two main lines of our results: (i) the mismatch in bias between the EVC and perceptual anisotropy and (ii) the strong correlation in variability between them, as illustrated in the revised Fig. 6. We want to emphasize that we’ve consistently stated this main claim throughout the manuscript, as reflected in the new Title (“**Bounded contribution of human early visual cortex to the topographic anisotropy in spatial extent perception**”), in the revised Abstract (“Our findings suggest that spatial extent perception builds on EVC’s spatial representation but requires an additional mechanism to transform its topographic bias”; lines 17-18), in one of the Results subsection title (“**The pRF anisotropy and the perceptual anisotropy differ in bias but share the inter-individual variability**”; lines 780-781) along with the revised Fig. 6 (, where (i) the mismatch and (ii) the link are jointly presented in a balanced way, and the inter-individual variability results are further analyzed and presented more thoroughly), and one of the Discussion subsection title (“**Bounded contribution of EVC to spatial extent perception**”; lines 1112-1184). We believe that this clears up confusion and provides a better presentation of our research.

Regarding the influences of radially and co-axiality on EVC’s pRF anisotropy, we clarify that we’ve now approached this issue in a more systematic way by identifying four scenarios for possible outcomes of the pRF anisotropy. We’ve illustrated these scenarios in the revised Fig. 1e,f, by distinguishing between “biasing” and “modulatory” influences. We’ve then described the outcomes of our fMRI experiment as the one corresponding to the scenario depicted in the right panel of Fig. 1f: EVC’s pRF anisotropy is radially biased, with a weak modulation by co-axiality, as reflected in the title of the Results section where our fMRI results were summarized (“**The pRF anisotropy of EVC is radially biased, with a weak modulation by co-axiality**”; line 273). We’ve further clarified the influences of the two factors on the EVC anisotropy by stating that the direction of its anisotropy “was biased by radially, and its magnitude was weakly modulated by co-axiality” in the first paragraph of the last Results subsection (lines 782-787) using the revised Fig. 6a. Lastly, based on this specific characterization of the EVC anisotropy, we discussed the aspects in which our findings are consistent with previous studies and

how we extended those studies (in the third paragraph of the Discussion subsection titled “**A fair share of the co-axial anisotropy in the spatial representation of EVC**”; lines 1012-1078).

[R2-C2] “Second, the finding that perceptual judgement of spatial extent is affected by the co-axial bias has already been described in a previous study which the current study builds on (Michel et al, 2013). This should be mentioned more clearly in the manuscript. Michel et al, 2013 also contains evidence against a radial perceptual bias, as the aspect ratio of non-oriented visual stimuli was judged veridically by human subjects despite the radial bias in EVC representations. Similarly, other studies that showed high-fidelity perception of circularity (Regan et al, 1992, 1996) already suggested that the physiological radial bias is discounted perceptually. These previous results should be properly mentioned and the way in which the current study expands on these results should be discussed.”

Our reply to [R2-C2] We thank the reviewer for bringing this issue to our attention. We agree with the reviewer's suggestion that we should properly mention the previous results on perceptual judgments of spatial extent and discuss how the current study expands on these findings. We've incorporated this suggestion in the revised manuscript, particularly in the Introduction and Discussion sections.

Particularly, we have mentioned Michel et al. (2013), along with other previous studies, throughout the manuscript, where necessary, to describe their findings and compare them with our results to highlight the similarities and differences between our study and theirs. In the Introduction section, specifically in the sixth paragraph, we acknowledged the rarity of studies that attempted to relate the topographic anisotropy in EVC to that in perception (lines 119-121). We've then stated our plan to expand on this work clearly. We've referred to Michel et al. (2013) thrice in the Discussion section. Firstly, we mentioned it along with two other imaging studies as previous work that demonstrated the influence of co-axiality on the EVC's representational anisotropy while discussing the co-axial anisotropy in the spatial representation of EVC. We've then clarified that our work expanded on these studies by “delineating the distinct natures and degrees of the influences of radially and co-axiality on the anisotropy in EVC” (in the third paragraph of the Discussion subsection titled “**A fair share of the co-axial anisotropy in the spatial representation of EVC**”; lines 1012-1078). Secondly, we focused on the work as previous work that demonstrated the influence of co-axiality on perceptual anisotropy. We've then clarified that our work confirms the co-axial bias and supports its robustness and generalizability and expanded on it by showing that radially also has a moderate modulatory influence on perceptual anisotropy (in the Discussion subsection titled “**Topographic anisotropy in human spatial extent perception**”; lines 1085-1095). Lastly, we mentioned the work along with other previous studies that have suggested the critical roles of EVC in spatial extent perception. We've then clarified that, while our finding of the shared inter-individual variability between EVC and perception supports the critical role of EVC to spatial extent perception, our finding of the mismatch in anisotropy bias between EVC and perception bounds its role (in the second paragraph of the Discussion subsection titled “**Bounded contribution of EVC to spatial extent perception**”; lines 1157-1162).

Although we agree with most of the points raised by the reviewer, we are puzzled by the statement that Michel et al. (2013) “contains evidence against a radial perceptual bias,” and Regan & Hamstra (1992) “suggested that the physiological radial bias is discounted perceptually.” We've thoroughly examined these studies several times and couldn't identify any findings that suggest, whether directly or indirectly, the presence or absence of radial influence on spatial extent perception. In our opinion, a fair

assessment of the influences of ‘radiality’ and ‘co-axiality’ on topographic anisotropy requires the manipulation of ‘stimulus orientation’ between radial vs tangential orientation at a given stimulus location in the polar space. Otherwise, it is difficult to evaluate the respective influences of these two anisotropy factors. In our revision, we’ve mentioned this point while referring to these two studies (in the second paragraph of the Discussion subsection titled “**Topographic anisotropy in human spatial extent perception**”; lines 1096-1107). Although we couldn’t further elaborate on this point in the manuscript, we’d like to share with the reviewer our reasoning behind our conclusion by providing further details below.

In one viewing condition of Michel et al. (2013), a pair of non-oriented plaid visual stimuli differing in the horizontal-to-vertical aspect ratio of their contrast envelop, was presented at $\pm 45^\circ$ intercardinal locations, and observers were asked to choose the one closer to a perfect circular shape. In this condition, observers consistently chose a visual stimulus with an aspect ratio of 1 rather than a vertically or horizontally elongated stimulus. However, these results do not provide conclusive evidence for the presence of the modulatory effect of radiality in perceived anisotropy. The perceived aspect ratio of “1” does not necessarily imply the absence of radial anisotropy because, even in the presence of radial anisotropy, the perceived aspect ratio of the plaid would remain 1 unless there is an imbalance between the co-axial anisotropies in the vertical and horizontal directions.

In the study conducted by Regan & Hamstra (1992), observers exhibited a high-fidelity perception of circularity in a task choosing the more vertically elongated stimulus from a pair of visual stimuli presented sequentially

Regan & Hamstra (1992), observers exhibited a high-fidelity perception of circularity in a task choosing the more vertically elongated stimulus from a pair of visual stimuli presented sequentially. However, the lack of information regarding stimulus location and eye fixation location in their paper makes it challenging to conclude that the radial bias is absent in perception. Suppose observers fixate their gaze at the center of the display and a visual stimulus is presented at the center of the display. It is difficult to evaluate the influence of radiality under this viewing condition because the contribution of radial bias to horizontal and vertical extension is balanced. Here we illustrate this point in the case of choosing a ‘vertically elongated one’ when a square stimulus is presented. This figure depicts radially biased spatial extension of the stimulus edges of the square representing the physically presented stimuli. We simply assumed that perceived spatial extent expands by ‘ α ’ along the radial direction (depicted by dark gray, magenta, and blue lines). This illustration demonstrates that the spatial extension of edges located at $y=0$ and $x=0$ determines perceived horizontal-to-vertical anisotropy. Specifically, edges located at the corners, such as those on the $y=x$ line, show the least increase of $\alpha/\sqrt{2}$ along the vertical and horizontal axes. Therefore, in the task of choosing a ‘vertically elongated one’, the spatial extension of edges located at $y=0$ and $x=0$ would determine horizontal-to-vertical anisotropy in the perceived spatial extents. Therefore, we reasoned that the high-fidelity perception of circularity in this study does not necessarily imply the absence of radial bias in perceived anisotropy. To summarize, in such a task, even in the presence of radial bias in perceptual anisotropy, the perceived horizontal-to-vertical aspect ratio – what observers estimated – remains 1 as long as the contribution of radially biased spatial extension to horizontal and vertical extension is balanced.

figure depicts radially biased spatial extension of the stimulus edges of the square representing the physically presented stimuli. We simply assumed that perceived spatial extent expands by ‘ α ’ along the radial direction (depicted by dark gray, magenta, and blue lines). This illustration demonstrates that the spatial extension of edges located at $y=0$ and $x=0$ determines perceived horizontal-to-vertical anisotropy. Specifically, edges located at the corners, such as those on the $y=x$ line, show the least increase of $\alpha/\sqrt{2}$ along the vertical and horizontal axes. Therefore, in the task of choosing a ‘vertically elongated one’, the spatial extension of edges located at $y=0$ and $x=0$ would determine horizontal-to-vertical anisotropy in the perceived spatial extents. Therefore, we reasoned that the high-fidelity perception of circularity in this study does not necessarily imply the absence of radial bias in perceived anisotropy. To summarize, in such a task, even in the presence of radial bias in perceptual anisotropy, the perceived horizontal-to-vertical aspect ratio – what observers estimated – remains 1 as long as the contribution of radially biased spatial extension to horizontal and vertical extension is balanced.

[R2-C3] “Third, some of the technical details are difficult to follow. For example, the analysis of the across-observer correlation in Fig. 6 is not explained clearly. What exactly is being correlated and what does this result mean? Similarly, the two analyses mentioned in lines 649-654 in the discussion are not clearly explained, motivated and interpreted. These analyses seem to be central to the conclusion of the paper and clarifying these issues will help in assessing the significance and novelty of the findings.”

Our reply to [R2-C3] We concur with the reviewer that “the analysis of the across-observer correlation in Fig. 6” and “the two analyses mentioned in lines 649-654 in the discussion” are crucial and thus should be presented clearly. We acknowledge that those analyses were poorly presented in the previously submitted manuscript. To improve the presentation, we substantially revised the manuscript, including the Introduction, Results, and Discussion sections, along with the associated figures.

Central to these revisions is a clear distinction between the ‘biasing’ and ‘modulatory’ influences of the two anisotropy factors (i.e., radially and co-axially) on the pRF and perceptual anisotropies. To overview the revisions, we (i) introduced that distinction from the outset of the revised manuscript (as illustrated in the revised Fig. 1e,f); (ii) described our findings based on this distinction made in (i) by specifying which factors are the ‘biasing’ or ‘modulatory’ factors and quantifying the degrees of the ‘biasing’ and ‘modulatory’ influences with refined indices for both EVC and perception in the Results section (as shown in Figs. 3,5 and the revised Fig. 6c); (iii) carried out two analyses to see how EVC relate to perception in topographic anisotropy of spatial representation based on (ii), one regarding the mismatch in anisotropy bias between EVC and perception (as shown in the revised Fig. 6a) and the other regarding the inter-individual co-variation between EVC and perception (as shown in the revised Fig. 6b,d); (iv) we discussed the contribution of EVC to spatial extent perception by jointly interpreting the outcomes of the two analyses in (iii) (as stated in the Discussion subsection titled “**Bounded contribution of EVC to spatial extent perception**”; lines 1112-1216).

As an important point of our revision, we’d like to bring the reviewer’s attention to the introduction of two new indices of anisotropy effects (lines 614-778 for **CI**; lines 811-847 for **CMI**). This has been introduced to demonstrate that the extent to which the modulatory influence of ‘co-axiality’ affects EVC’s pRF anisotropy, as captured by ‘co-axial modulation index (**CMI**),’ is related to the degree of co-axial bias in spatial extent perception, as captured by ‘co-axial bias index (**CI**).’ By using these two indices, we were able to show that there is a significant correlation between the inter-individual difference in CMI of EVC and that in CI of perception, as depicted in the revised Fig. 6d. This finding, along with the strong inter-individual correlation between the radial bias in EVC and the modulatory influence of radially in perception, as shown in the revised Fig. 6b, suggests that the variabilities in both the biasing influence of radially and the modulatory influence of co-axiality in EVC respectively propagate to the corresponding types of variability in perception.

[Specific comments]

[R2-C4] “In lines 480-481, the authors write: “perceived spatial extent must be subject to the topographic bias in neural properties”. I think that this statement needs to be qualified. It is important for the authors to distinguish between static anisotropies, such as the radial bias, and stimulus-dependent anisotropies, such as the co-axial bias. If the EVC hypothesis is true, decoding circuits that receive information from EVC must take into account such static anisotropies. The naïve hypothesis that there would be a simple

mapping between spread of activity in EVC and spatial extent perception that ignores such static anisotropies and distortions is not tenable and can easily be rejected. A more realistic test of the EVC hypothesis is whether stimulus-specific physiological biases such as the axial bias in EVC correlate with perception, and the results of the current study, which confirm the findings of Michel et al 2013, demonstrate that such perceptual biases exist and are robust, providing support to the EVC hypothesis. As discussed above, these issues need to be explained more clearly throughout the manuscript.”

[Reply to R2-C4] Since this issue aligns with one of the previous comments, **[R2-C1]**, and two of the other reviewer’s comments, **[R1-C1]** and **[R1-C2]**, it has been effectively addressed in the revisions we made in response to those comments. As mentioned in our replies to these comments, we’ve realigned our study’s aim and reframed the manuscript’s storyline. We’ve done this without relying on the narrowly (naïvely) defined EVC hypothesis. Thus, any sentences or phrases that presumed the narrowly defined EVC hypothesis have been removed. Instead, with this revised aim and storyline, we’ve presented our work as an attempt to refine the current understanding of EVC’s contribution to spatial extent perception by putting it in the context of topographic anisotropy. Eventually, we’ve derived our main claim that EVC makes a critical and limited contribution to spatial extent perception by interpreting together the two main lines of our findings, as described earlier.

[R2-C5] “While the co-axial bias effects in EVC are weak and may not reach statistical significance in the current study which uses fMRI, it is important to mention that these effects are robust in macaque V1 when measured with optical imaging (Michele et al, 2013).”

Our reply to [R2-C5] As we mentioned in our reply to **[R2-C2]**, we’ve emphasized this work’s finding regarding the co-axial anisotropy in the spatial representation of EVC. In doing so, we’ve also clarified that we also found a weak modulatory influence of co-axiality on EVC’s pRF anisotropy although it failed to reach a significant level, as stated in the third paragraph of the Discussion section titled “**A fair share of the co-axial anisotropy in the spatial representation of EVC**” (lines 1012-1078).

[R2-C6] “Lines 642-643: “co-axial bias may exist but is not as strong as previously thought”. The authors should need to be more explicit and explain which expectations they are referring to.”

Our reply to [R2-C6] We acknowledge that our use of the phrase "as previously thought" may have caused confusion and ambiguity. Therefore, we have revised the sentence to make it clearer: "Our findings suggest that the nature and strength of the co-axial anisotropy in human EVC may need to be re-characterized: its influence on spatial representation may be modulatory in nature and weak in strength." (in the last sentence of the last paragraph of the Discussion subsection titled “**A fair share of the co-axial anisotropy in the spatial representation of EVC**”; lines 1080-1082).

[R2-C7] “Was the analysis of the pRF restricted to a single eccentricity as in Fig. 3A? If so, why? If not, how were the results combined across eccentricities given that the size of the pRF changes with eccentricity?”

Our reply to [R2-C7] To clarify, we analyzed pRF anisotropy across the entire range of visual stimulation eccentricities, from 0.7° to 7.9°. In Fig. 3, we schematically presented the mean pRF extent, the anisotropy of which was obtained by averaging the anisotropies across entire voxels, at a representative eccentricity on the horizontal axis for illustrative purposes. Prompted by the reviewer's query, we've provided the anisotropy data across the full range of eccentricities for the two stimulus orientation conditions for each area of EVC in Supplementary Figure 2. This figure is referred to in the last paragraph of the Results subsection titled "**The pRF anisotropy of EVC is radially biased, with a weak modulation by co-axiality**".

REVIEWERS' COMMENTS:

Reviewer #1 (Remarks to the Author):

In this revision, the authors have reframed their manuscript to explain their experiments and results more clearly, and I wanted to thank the authors for addressing my comments so thoroughly. This version lays out the possible outcomes very clearly and presents a more nuanced picture of their findings. I spotted a couple of minor things that should be changed:

1. In a couple of places, the authors indicate that the coaxial bias weakly modulates the pRF anisotropy (e.g., the heading on line 273, line 861). Since the effects here are not statistically significant, I would just be a bit clearer in indicating that there's not enough evidence to support this coaxial effect.
2. Figure 1 caption (line 1810): this should say "co-axiality (a) and radiality (b)"

Reviewer #2 (Remarks to the Author):

The paper is much improved and the authors should be commended for their effort to address the reviewers' comments. I still have a few suggestions, but I leave it to the authors to decide whether or not to address those in the final version of the manuscript.

While the authors have made an effort to de-emphasize the narrowly defined early visual cortex (EVC) hypothesis, this hypothesis still seems to play a major role in the framing of the work. I still think that the hypothesis that the spatial extent of a visual stimulus can be directly read out from the spread of activity in EVC (without compensating for distortions in the retinotopic map and in the size and shape of the pRF) is a strawman hypothesis that can easily be rejected. It is well known that due to anisotropies in the retinotopic map, a circular stimulus produces a spread of activity that is highly non-circular in EVC irrespective of the stimulus' texture, and that the size of a small stimulus cannot be estimated from the spread of activity in EVC without taking into account the size of the cortical point image. Therefore, given that our perception of shape and size is veridical for most stimuli, if the spread of activity in EVC contributes to shape and size perception, downstream circuits that mediate perception must take these factors into account. This study provides support for the hypothesis that the spread of activity in EVC contributes to shape and size perception and that the compensation for some of these distortions occur downstream to EVC.

I also think that it important to distinguish between static anisotropies that depend only on the position of the stimulus in the visual field (such as the radial bias in the retinotopic map), and anisotropies that are stimulus-dependent, such as the co-axial bias. While static anisotropies are relatively easy for downstream circuits to compensate for, stimulus-dependent anisotropies are more challenging for downstream circuits to compensate for. This could explain why the radial bias, which is large physiologically, has a small perceptual effect, while the co-axial bias, which is small physiologically, has a larger perceptual effect. I did not see this issue addressed clearly in the revised version.

The anisotropy in the spread of activity in EVC is determined by the anisotropy in the pRF and by the anisotropy in the cortical magnification factor. As far as I can tell, the paper focuses exclusively on anisotropies in the pRF but does not mention anisotropies in the retinotopic map itself (i.e., the cortical magnification factor being different in the radial and tangential directions). Clarifying the possible contribution of anisotropies in the cortical magnification factor to the physiological and perceptual results could strengthen the paper.